# Spatial Structure and Selective Text Jointly Facilitate Image Clustering

**Zizheng Jiu**[1,2,3†]**, Feijiang Li**[1,2,3†]**, Jieting Wang**[1,2,3]**, Yuhua Qian**[1,2,3*]**, Lu Chen**[1,2,3]
[1]Institute of Big Data Science and Industry, Shanxi University
[2]Key Laboratory of Evolutionary Science Intelligence of Shanxi Province, Taiyuan. Shanxi. China
[3]School of Artificial Intelligence, Shanxi University
`jinchengqyh@126.com`

## Abstract

Image clustering is a fundamental task in visual machine learning. A key research direction in this field is the incorporation of prior knowledge. Recently, such prior knowledge has evolved from internal compactness constraints to external textual guidance. In particular, the introduction of textual modalities through CLIP has demonstrated impressive performance. However, CLIP is designed primarily for image–text alignment and may not be sufficient to capture clustering structures. Moreover, existing approaches often assume that textual features are universally beneficial, overlooking their varying suitability for different datasets. To address these issues, we propose using spatial structure and selective text jointly to facilitate image clustering (SATC). Specifically, we design a graph attention network (GAT)-based encoder to capture relational dependencies among image patches, thereby extracting spatial features to facilitate clustering. In addition, we introduce a textual feature selector that uses the potential clustering compactness of textual features as the selection criterion and adaptively integrates them into the clustering process. Theoretical guidance is provided for this selector. Finally, the cluster assignment is produced through tri-modal mutual distillation. Extensive experiments on 18 benchmark datasets demonstrate the effectiveness of SATC. The experimental results further verify the rationality of the textual feature selector. The project page is available at https://zizhjiu.github.io/SATC/.

## 1 Introduction

Image clustering aims to group unlabeled images into semantically meaningful clusters, playing a key role in various real-world applications, such as image retrieval and dataset organization (Huang et al., 2024). A core aspect of image clustering lies in the incorporation and effective use of prior knowledge to reveal the underlying data structure. In the absence of explicit labels, prior knowledge provides essential guidance to the clustering process.

Most traditional and deep clustering methods rely on the prior of cluster compactness—assuming that samples belonging to the same category naturally cluster together in the feature space. Classical methods, such as K-Means (Krishna & Murty, 1999), operate on handcrafted features. These shallow features fail to capture complex visual differences. Deep clustering methods overcome this limitation by jointly learning feature representations and cluster assignments in an end-to-end manner. For instance, DEC (Xie et al., 2016) optimizes a self-supervised objective to refine clusters while optimizing feature representations. Subsequent works (Han et al., 2020; Yu et al., 2020) further enhance clustering performance by improving representation quality and cluster discrimination. Despite these advances, such methods remain constrained to internal supervision signals derived solely from the data itself.

---

† These authors contribute equally to this work.    * Corresponding author.

Recently, inspired by cross-modal foundation models such as CLIP (Radford et al., 2021), researchers have begun to explore the use of additional textual guidance as prior knowledge for image clustering. For instance, TAC (Li et al., 2023) proposes leveraging rich external textual knowledge to guide the clustering process, introducing a new direction for image clustering. Built upon CLIP, TAC exploits the natural semantic alignment between image–text pairs to enhance clustering quality. By enabling cross-modal learning, TAC introduces more informative prior knowledge into the clustering framework, illustrating the potential of external textual priors to overcome the limitations inherent in methods that rely solely on internal supervision signals.

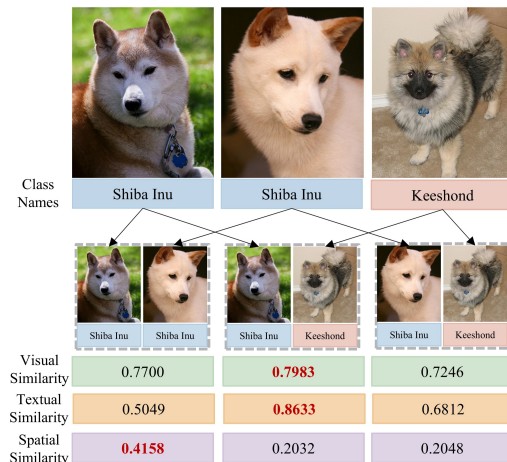

Figure 1: Pairwise cosine similarities among two Shiba Inu images and one Keeshond image from the OxfordPets dataset, evaluated across visual, textual, and spatial modalities. Visual features are extracted using CLIP, textual features are obtained following TAC (Li et al., 2023), and spatial features are derived from our proposed SATC.

However, CLIP is primarily trained to align images and text within a shared semantic space, which may compromise the representation of spatial structures. The internal compactness of the image is influenced by the spatial structure, which encodes rich information such as an object's shape, position, and part layout. This information plays a key role in downstream clustering performance. As illustrated in Figure 1, when clustering two Shiba Inu images and one Keeshond image, both visual and textual similarities mistakenly favor aligning a Shiba Inu with the Keeshond rather than correctly grouping the two Shiba Inu images. Compared to visual and textual modalities, spatial modalities more effectively capture intra-class similarity and better distinguish between categories.

Moreover, textual features are often assumed to be universally beneficial across different datasets, without assessing their suitability to specific data. In practice, textual features may not always be suitable for all datasets (Zhu et al., 2025a). As illustrated in Figure 1, textual similarities can sometimes fail to capture category-level distinctions, and indiscriminately incorporating them may introduce noise, ultimately degrading clustering performance. We evaluated the TAC method across all 18 datasets listed in Table 4 in Appendix A and compared its performance against a text-free baseline. The results (see Appendix Table 8) demonstrate that textual features do not universally improve clustering performance across all datasets.

To address these challenges, we propose a novel image clustering framework, Spatial Structure and Selective Text Jointly Facilitate Image Clustering (SATC). SATC integrates three key components to improve clustering performance. First, it employs a GAT-based spatial encoder to capture spatial relationships among image patches, thereby overcoming the limitations of CLIP in capturing local structure. Second, it introduces a textual feature selector that evaluates the potential clustering compactness of textual features to selectively incorporate the textual feature into the clustering process, reducing the misleading of textual signals. Third, it adopts a tri-modal mutual distillation strategy to improve cluster discrimination by jointly leveraging visual, spatial, and textual modalities.

The main contributions of the paper are as follows:

- We propose a textual feature selector that evaluates and adaptively incorporates textual information based on its potential clustering compactness, which is used to improve clustering robustness across diverse datasets. Theoretical guidance for this selector is provided.

- We introduce a spatial feature that explicitly encodes relational structure among image patches and further integrate it with visual features and beneficial textual features through a tri-modal mutual distillation framework, ultimately generating cluster assignments.

- We conduct extensive experiments on 18 benchmark datasets, demonstrating that our method outperforms state-of-the-art approaches and validating the effectiveness of combining spatial features and beneficial textual guidance for image clustering.

## 2 RELATED WORK

### 2.1 IMAGE CLUSTERING

**Internal Supervision Signal Methods.** Most of the existing methods rely on an internal compactness constraint to guide the learning of clustering-friendly representations. They typically assume that samples from the same category are naturally closer in the feature space and optimize the representations accordingly. Early works such as DEC (Xie et al., 2016) refine cluster assignments by minimizing the KL divergence between current predictions and a sharpened target distribution. Methods like IIC (Ji et al., 2019), SCAN (Van Gansbeke et al., 2020), and CC (Li et al., 2021) leverage prediction consistency under strong data augmentations to facilitate unsupervised clustering, whereas approaches such as DeepCluster (Caron et al., 2018) and SPICE (Niu et al., 2022) iteratively assign pseudo-labels to supervise feature learning. Graph-based extensions like GATCluster (Niu et al., 2020) introduce graph attention mechanisms to model neighborhood dependencies and improve clustering quality.

**External Textual Guidance Methods.** Recently, some methods rely on external textual prior knowledge to guide the learning of clustering-friendly representations. They aim to overcome the limitations of internal supervision by introducing cross-modal supervision from the textual modalities. SIC (Cai et al., 2023) generates pseudo-labels in the textual feature space, while TAC (Li et al., 2023) utilizes textual features generated by CLIP as auxiliary supervision to enhance clustering performance. These methods highlight the potential of external textual guidance. However, they often rely on the assumption that the textual features are consistently reliable.

### 2.2 CLIP-BASED REPRESENTATION LEARNING

Contrastive Language-Image Pretraining (CLIP) (Radford et al., 2021) has emerged as a powerful foundation model by aligning visual and textual features through large-scale contrastive learning. An increasing number of studies have leveraged CLIP across various downstream tasks. For example, WeakCLIP (Zhu et al., 2025b) adapts CLIP to weakly supervised semantic segmentation. In the context of few-shot learning, MORN (Ni et al., 2024) utilizes CLIP to generate modality-specific prototypes and integrates them via a cross-modal enhancement module to improve action recognition. Similarly, CLIP-RPN (Guan & Yoshie, 2025) exploits CLIP's semantic perception to adaptively route features for image deraining based on rain-pattern awareness. These diverse applications demonstrate CLIP's versatility in leveraging cross-modal representations. However, although CLIP excels at image-text alignment, it lacks explicit mechanisms for capturing relational dependencies among image patches—a limitation that constrains its effectiveness for image clustering tasks.

To address the limitations, we propose a GAT-based spatial encoder that explicitly captures relational dependencies among image patches, alongside a textual feature selector that assesses and adaptively integrates textual modality across different datasets. Building on these components, we develop SATC—a robust and adaptive image clustering framework that effectively leverages both spatial structures and textual priors.

## 3 THE PROPOSED METHOD

In this section, we describe the proposed SATC, illustrated in Figure 2. SATC is designed to overcome the limitation of CLIP while mitigating the impact of unreliable textual feature guidance by incorporating (a) visual and spatial feature extraction, (b) compactness-aware textual feature selection, and (c) tri-modal mutual distillation.

### 3.1 VISUAL AND SPATIAL FEATURE EXTRACTION

For the raw image dataset $\mathcal{D} = \{x_n\}_{n=1}^{N}$, we leverage the pretrained CLIP model to extract visual features $z^{\text{visual}} \in \mathbb{R}^d$ for images in dataset $\mathcal{D}$, where $d$ is the feature dimension predefined by the CLIP model.

Motivated by the work of (Qian et al., 2015), which introduces a space-structure-based representation to improve clustering for categorical data, we explore extracting spatial structure features to

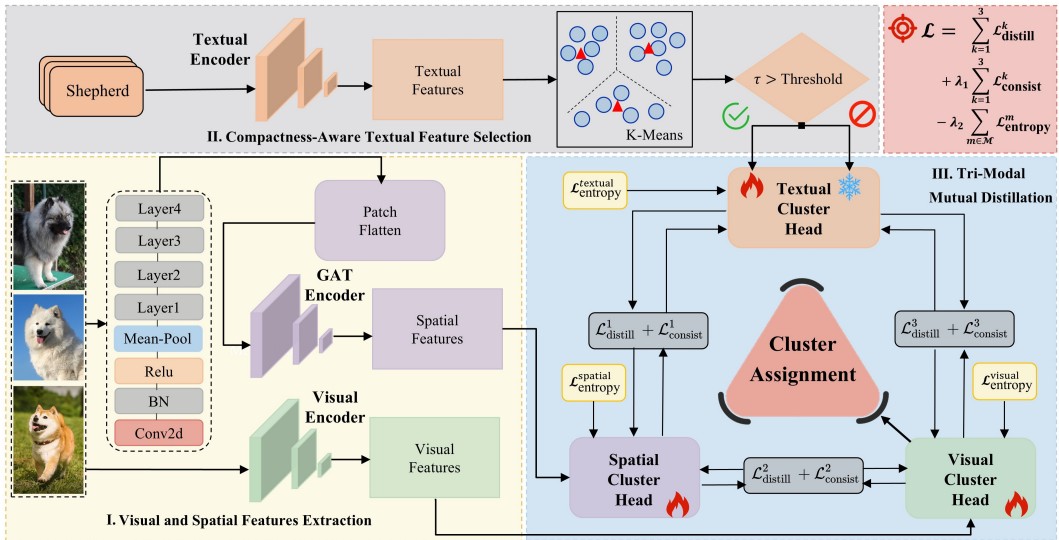

Figure 2: Overview of the SATC. The pseudocode is presented in Appendix H Algorithm 1.

facilitate improved image clustering performance. To this end, we apply a Graph Attention Network (GAT) (Veličković et al., 2017) to patches within each individual image to capture spatial relationships among different regions of the same image. GAT is a neural network designed for graph-structured data that updates each node by attending to its neighbors using learnable attention weights. The attention mechanism in GAT enables adaptive weighting of neighboring patches, allowing it to effectively capture relationships among image patches. Specifically, for each image, we first divide it into patches and extract patch-level feature using a pretrained ResNet-50 (Radford et al., 2021) with the final pooling and fully connected layers removed. Each patch feature serves as a graph node, and edges are constructed by connecting semantically related patches within the same image. The resulting node features are denoted as $X = [x_1, \ldots, x_M]^\top \in \mathbb{R}^{M \times d}$, where $M$ is the number of patches in the image, and the edge set is $E = \{(i,j) \mid i \neq j, \ i,j \in \{1, \ldots, M\}\}$ based on feature similarity.

Then, we feed the node features $X$ along with the edge set $E$ into GAT, which updates each node by aggregating information from its neighbors with attention. The updated representation $x_i'$ for node $i$ can be expressed as

$$x_i' = \sigma \left( \sum_{j \in \mathcal{N}(i)} \frac{\exp(f(x_i, x_j))}{\sum_{k \in \mathcal{N}(i)} \exp(f(x_i, x_k))} \, x_j \right), \tag{1}$$

where $\mathcal{N}(i)$ denotes the set of neighbors of node $i$, $\sigma(\cdot)$ is a nonlinear activation function, and $f(x_i, x_j)$ is a learnable function that computes the attention score between nodes $i$ and $j$ in GAT.

After obtaining the refined node features $X' = [x_1', \ldots, x_M']^\top \in \mathbb{R}^{M \times d}$, we apply global average pooling over the $M$ nodes, obtaining the spatial embeddings $z^{\text{spatial}} \in \mathbb{R}^d$. The embeddings are then used as an additional input to the downstream clustering process.

## 3.2 COMPACTNESS-AWARE TEXTUAL FEATURE SELECTION

### 3.2.1 TEXTUAL FEATURE EXTRACTION

To generate textual features for clustering, we adopt the text counterpart construction process introduced in TAC (Li et al., 2023). Specifically, we first collect candidate nouns $T_j \in \mathbb{R}^d$ from WordNet (Miller, 1995) using CLIP.

To align textual features with the visual features, we first cluster the visual features $z^{\text{visual}}$ into $K_1 = \lfloor N/300 \rfloor$ clusters by K-Means. The resulting cluster centers are denoted $\{\mu_l\}_{l=1}^{K_1}$. To ensure that each semantic center is represented by highly discriminative textual concepts, we identify the top-

5 confident nouns $\{\bar{T}_m\}_{m=1}^5$ for each cluster. These are nouns that exhibit the highest posterior probabilities of belonging to the corresponding cluster center. Formally, the posterior probability of assigning a noun $T_j$ to a cluster $l$ is computed as:

$$p(y = l \mid T_j) = \frac{e^{(\mathrm{sim}(T_j, \mu_l))}}{\sum_{i=1}^{K1} e^{(\mathrm{sim}(T_j, \mu_i))}}, \tag{2}$$

where $\mathrm{sim}(\cdot, \cdot)$ denotes the cosine similarity.

The textual feature $z_i^{\text{textual}}$ corresponding to the visual feature $z_i^{\text{visual}}$ (for sample $i \in \{1, \dots, N\}$) is computed via a soft retrieval over $\bar{T}_m$:

$$z_i^{\text{textual}} = \sum_{j=1}^{5} \frac{e^{(\mathrm{sim}(z_i^{\text{visual}}, \bar{T}_j)/\beta_1)}}{\sum_{h=1}^{5} e^{(\mathrm{sim}(z_i^{\text{visual}}, \bar{T}_h)/\beta_1)}} \cdot \bar{T}_j. \tag{3}$$

where $\beta_1 = 0.005$ is a temperature hyperparameter.

### 3.2.2 TEXTUAL FEATURE SELECTOR

To leverage the textual modality that is beneficial for image clustering, we introduce a textual feature selector based on textual compactness. Inspired by the concept of intra-cluster compactness in internal supervision signal clustering methods. Theoretically, if features belonging to the same class are highly concentrated in the vector space, they are more discriminative and can effectively distinguish different categories. For the textual modality, a lower compactness score indicates that textual features are highly clustered and may be semantically redundant, whereas a higher score reflects greater semantic diversity, suggesting that the textual modality can provide more valuable guidance for clustering. This motivates the use of textual compactness $\tau$ as a criterion.

We first generate the textual features and assess their compactness $\tau$ by clustering the pre-extracted textual embeddings $z^{\text{textual}}$ into $K_t$ clusters using K-Means, where $K_t$ denotes the number of textual prototypes. We set $K_t = 10$ for all datasets in experiments. For the $j$-th cluster, the assigned textual samples form the set $D_j$, and the cluster center $C_j$ is computed as:

$$C_j = \frac{1}{|D_j|} \sum_{z_j^{\text{textual}} \in D_j} z_j^{\text{textual}}, \tag{4}$$

Based on the $C_j$, the textual compactness metric is defined as the average intra-cluster distance between the textual features and their corresponding cluster centers:

$$\tau = \frac{1}{K_t} \sum_{j=1}^{K_t} \frac{1}{|D_j|} \sum_{z_j^{\text{textual}} \in D_j} \|z_j^{\text{textual}} - C_j\|_2. \tag{5}$$

The textual compactness $\tau$ is used as a dataset-level prior. Before training the tri-modal distillation framework, we compute $\tau$ on the entire dataset's textual feature. If $\tau$ is higher than a threshold, we consider the textual modality to be beneficial and include it in the mutual distillation process; otherwise, we exclude it. In Appendix L, a theoretical guidance for utilizing textual compactness is given. This adaptive selection ensures that the textual modality contributes positively only when it provides sufficiently diverse and discriminative information. The effectiveness of this textual compactness-based selection strategy is validated through extensive experiments in Section 4.3.1.

### 3.3 TRI-MODAL MUTUAL DISTILLATION

To effectively integrate tri-modal information for image clustering, we propose a tri-modal distillation framework. This framework takes pre-trained visual ($z^{\text{visual}}$), spatial ($z^{\text{spatial}}$), and textual ($z^{\text{textual}}$; used only when $\tau >$ threshold) features as input. Each modality feature is projected into a cluster assignment distribution via a dedicated MLP cluster head. Following mutual distillation, spatial and textual feature serve as auxiliary information in the image clustering task, and therefore we take the cluster assignment distribution from the distilled visual modality cluster head as the final assignment. A detailed comparison of different modality cluster heads is provided in Appendix K.

The total training objective integrates all components with balancing hyperparameters $\lambda_1$ and $\lambda_2$:

$$\mathcal{L} = \sum_{k=1}^{3} \mathcal{L}_{\text{distill}}^{k} + \lambda_1 \sum_{k=1}^{3} \mathcal{L}_{\text{consist}}^{k} - \lambda_2 \sum_{m \in \mathcal{M}} \mathcal{L}_{\text{entropy}}^{m}. \tag{6}$$

where $\mathcal{M} = \{\text{visual, textual, spatial}\}$. There are two cross-modal losses, which are distillation loss and consistency loss, and one internal loss that is evaluated by entropy loss. The three loss functions act synergistically, enabling the model to achieve better performance. A detailed analysis of each loss function, both individually and in combination, is provided in Appendix E. Specifically, the losses are:

- $\mathcal{L}_{\text{distill}}^{1}$, $\mathcal{L}_{\text{consist}}^{1}$: the distillation and consistency losses between visual and spatial modalities;
- $\mathcal{L}_{\text{distill}}^{2}$, $\mathcal{L}_{\text{consist}}^{2}$: the distillation consistency losses between textual and spatial modalities;
- $\mathcal{L}_{\text{distill}}^{3}$, $\mathcal{L}_{\text{consist}}^{3}$: the distillation and consistency losses between visual and textual modalities;
- $\mathcal{L}_{\text{entropy}}^{\text{visual}}$, $\mathcal{L}_{\text{entropy}}^{\text{textual}}$, $\mathcal{L}_{\text{entropy}}^{\text{spatial}}$: the entropy loss of the visual, textual, and spatial modalities.

To facilitate understanding, we detail the computation of $\mathcal{L}_{\text{distill}}^{1}$, $\mathcal{L}_{\text{consist}}^{1}$, and $\mathcal{L}_{\text{entropy}}^{\text{visual}}$ as representative examples, the remaining losses are computed analogously.

**Mutual Distillation Loss.** Each loss encourages the cluster assignment distributions of two modalities to be aligned. Taking the visual and spatial modalities for example, $c_i^{\text{visual}}$ and $c_i^{\text{spatial}}$ denote the soft cluster assignments of the $i$-th sample from the two modalities, the distillation loss is:

$$\mathcal{L}_{\text{distill}} = \frac{1}{N} \sum_{i=1}^{N} \left( \mathcal{L}_i^{S \to V} + \mathcal{L}_i^{V \to S} \right), \tag{7}$$

$$\mathcal{L}_i^{S \to V} = -\log \frac{\exp\left(\text{sim}(c_i^{\text{visual}}, c_i^{\text{spatial}})/T\right)}{\sum_{j=1}^{N} \exp\left(\text{sim}(c_i^{\text{visual}}, c_j^{\text{spatial}})/T\right)}, \tag{8}$$

$$\mathcal{L}_i^{V \to S} = -\log \frac{\exp\left(\text{sim}(c_i^{\text{spatial}}, c_i^{\text{visual}})/T\right)}{\sum_{j=1}^{N} \exp\left(\text{sim}(c_i^{\text{spatial}}, c_j^{\text{visual}})/T\right)}. \tag{9}$$

where $T$ is a temperature hyperparameter.

**Consistency Loss.** To further enhance alignment between modalities, we introduce a consistency loss that encourages similar cluster assignments between each anchor sample and its other modality. Taking the visual and spatial modalities for example, the loss is computed as:

$$\mathcal{L}_{\text{consist}}^{1} = -\frac{1}{N} \sum_{i=1}^{N} \log \left( \left(c_i^{\text{visual}}\right)^{\top} c_i^{\text{spatial}} \right) \tag{10}$$

**Entropy Loss.** To prevent degenerate solutions and encourage confident cluster assignments, we compute the entropy of each sample's cluster assignment distribution and take the average over all samples. The entropy loss for the visual modality is defined as:

$$\mathcal{L}_{\text{entropy}}^{\text{visual}} = -\frac{1}{N} \sum_{i=1}^{N} \frac{1}{K} \sum_{j=1}^{K} c_{i,j}^{\text{visual}} \log c_{i,j}^{\text{visual}}, \tag{11}$$

where $c_{i,j}^{\text{visual}}$ denotes the soft assignment probability of the $i$-th sample to the $j$-th cluster in visual modality and $K$ is the number of clusters.

## 4 EXPERIMENTAL ANALYSIS

In this section, we conduct comparative experiments and ablation studies to investigate the effectiveness and robustness of SATC.

## 4.1 EXPERIMENTAL SETTINGS

**Datasets and Evaluation Metric.** The experimental analyses are conducted on 18 diverse benchmark datasets. Detailed descriptions about these datasets are provided in Appendix A. All experiments use the original train-test splits. To assess performance, we employ three standard metrics, which are Accuracy (ACC), Normalized Mutual Information (NMI) (Estévez et al., 2009), and Adjusted Rand Index (ARI) (Steinley, 2004), where higher values indicate better clustering performance. To ensure statistical reliability, we report the mean results over ten independent runs with different random seeds.

**Implementation Details.** We use the CLIP ViT-B/32 model (Dosovitskiy et al., 2020) for visual and textual feature extraction. We process WordNet nouns (Miller, 1995) using CLIP to generate candidate nouns. For clustering, we adopt three modality-specific MLP cluster heads (512-512-$K$). The model is trained for 200 epochs with a batch size of 512. We employ an early stopping mechanism that terminates the distillation process when the total loss fails to decrease for 10 iterations. According to TAC (Li et al., 2023), we set the temperature for mutual distillation to $T = 0.5$. Based on extensive experiments, we set the weights for the consistency and entropy losses to $\lambda_1 = 1.0$ and $\lambda_2 = 5.0$, respectively (Section 4.4), and the compactness threshold to 0.33 (Section 4.3.1).

## 4.2 COMPARISON EXPERIMENT

We evaluate SATC on 5 widely used datasets, including ImageNet-10, ImageNet-Dogs, STL-10, CIFAR-10, and CIFAR-100. To assess its effectiveness, we compare it with a wide range of classical and state-of-the-art clustering methods, including K-Means (Krishna & Murty, 1999), DEC (Xie et al., 2016), DAC (Chang et al., 2017), DCCM (Wu et al., 2019), DSEC (Chang et al., 2018), GATCluster (Niu et al., 2020), PLCA (Huang et al., 2020), CC (Li et al., 2021), C3 (Sadeghi et al., 2022), MICE (Tsai et al., 2020), IDFD (Tao et al., 2021), TCL (Li et al., 2022), CONCUR (Deshmukh et al., 2022), SPICE (Niu et al., 2022), DPAC (Yan et al., 2024), TAC (Li et al., 2023), DINOv2-VitB/14 (Oquab et al., 2023) + K-Means, DINOv3-VitB/16 (Siméoni et al., 2025) + K-Means, Turtle (Gadetsky et al., 2024), GradNorm (Peng et al., 2025), and LFSS (Li et al., 2025).

Table 1: Clustering performance of different approaches evaluated by Accuracy (ACC%), Normalized Mutual Information (NMI%), and Adjusted Rand Index (ARI%). "–" indicates unavailable results. The best and second-best results are highlighted in **bold** and underline, respectively.

| Method | ImageNet-10 | | | ImageNet-Dogs | | | STL-10 | | | CIFAR-10 | | | CIFAR-100 | | |
|---|---|---|---|---|---|---|---|---|---|---|---|---|---|---|---|
| | ACC | NMI | ARI | ACC | NMI | ARI | ACC | NMI | ARI | ACC | NMI | ARI | ACC | NMI | ARI |
| K-Means | 24.1 | 11.9 | 5.7 | 10.5 | 5.5 | 2.0 | 19.2 | 12.5 | 6.1 | 22.9 | 8.7 | 4.9 | 13.0 | 8.4 | 2.8 |
| DEC | 38.1 | 28.2 | 20.3 | 19.5 | 12.2 | 7.9 | 35.9 | 27.6 | 18.6 | 30.1 | 25.0 | 16.1 | 18.5 | 13.6 | 12.5 |
| DAC | 52.7 | 39.4 | 30.2 | 27.5 | 21.9 | 11.1 | 47.0 | 36.6 | 25.7 | 52.2 | 40.0 | 30.1 | 23.8 | 18.5 | 8.8 |
| DCCM | 71.0 | 60.8 | 55.5 | 38.3 | 32.1 | 18.2 | 48.2 | 37.6 | 26.2 | 62.3 | 49.6 | 40.8 | 32.7 | 28.5 | 17.3 |
| DSEC | 67.4 | 58.3 | 52.2 | 26.4 | 23.6 | 12.4 | 48.2 | 40.3 | 28.6 | 47.8 | 43.8 | 34.0 | 25.5 | 21.2 | 11.0 |
| GATCluster | 76.2 | 60.9 | 57.2 | 33.3 | 32.2 | 20.0 | 58.3 | 44.6 | 36.3 | 61.0 | 47.5 | 40.2 | 28.1 | 21.5 | 11.6 |
| PICA | 87.0 | 80.2 | 76.1 | 35.2 | 35.2 | 20.1 | 71.3 | 61.1 | 53.1 | 69.6 | 59.1 | 51.2 | 33.7 | 31.0 | 17.1 |
| CC | 89.3 | 85.9 | 82.2 | 42.9 | 44.5 | 27.4 | 85.0 | 76.4 | 72.6 | 79.0 | 70.5 | 63.7 | 42.9 | 43.1 | 26.6 |
| C3 | 94.2 | 90.5 | 86.1 | 43.4 | 44.8 | 28.0 | – | – | – | 83.8 | 74.8 | 70.7 | 45.1 | 43.4 | 27.5 |
| MiCE | – | – | – | 43.9 | 42.3 | 28.6 | 75.2 | 63.5 | 57.5 | 83.5 | 73.7 | 69.8 | 44.0 | 43.6 | 28.0 |
| IDFD | 95.4 | 89.8 | 90.1 | 59.1 | 54.6 | 41.3 | 75.6 | 64.3 | 57.5 | 81.5 | 71.1 | 66.3 | 42.5 | 42.6 | 26.4 |
| TCL | 89.5 | 87.5 | 83.7 | 64.4 | 62.3 | 51.6 | 86.8 | 79.9 | 75.7 | 88.7 | 81.9 | 78.0 | 53.1 | 52.9 | 35.7 |
| ConCUR | 95.8 | 90.7 | 90.9 | 69.5 | 63.0 | 53.1 | 74.9 | 63.6 | 56.6 | 84.6 | 76.2 | 71.5 | 47.9 | 46.8 | 30.3 |
| SPICE | 96.9 | 92.7 | 93.3 | 67.5 | 62.7 | 52.6 | 92.9 | 86.0 | 86.5 | 91.8 | 85.0 | 83.6 | 58.4 | 58.3 | 42.2 |
| DPAC | 97.0 | 92.5 | 93.5 | 72.6 | 66.7 | 59.8 | 93.4 | 86.3 | 86.1 | 93.4 | 87.0 | 86.6 | 55.5 | 54.2 | 39.3 |
| DINOv2 | 96.3 | 93.7 | 91.6 | 87.5 | 88.1 | 80.4 | 77.4 | 81.5 | 64.0 | 77.0 | 81.6 | 64.5 | 62.5 | 69.8 | 46.1 |
| DINOv3 | 96.7 | 93.8 | 92.5 | 48.3 | 48.2 | 27.3 | 98.1 | 96.0 | 96.0 | 90.1 | 85.0 | 81.1 | 57.6 | 70.3 | 42.6 |
| Turtle | 99.3 | 98.3 | 98.6 | 49.6 | 44.6 | 32.4 | 98.4 | 95.5 | 96.1 | 86.6 | 78.8 | 75.4 | 46.4 | 58.7 | 34.5 |
| GradNorm | 99.4 | 98.7 | 98.7 | 81.2 | 81.0 | 70.9 | 98.3 | 95.6 | 96.2 | 91.1 | 82.6 | 81.5 | – | – | – |
| LFSS | 93.2 | 85.6 | 85.7 | 69.1 | 61.7 | 53.3 | 86.1 | 77.1 | 74.0 | 93.4 | 87.2 | 86.6 | – | – | – |
| TAC | 99.5 | 98.5 | 98.8 | 84.4 | 77.4 | 72.0 | 98.3 | 95.7 | 96.3 | 92.2 | 83.7 | 83.6 | 59.0 | 68.6 | 42.7 |
| **SATC** | **99.8** | **99.2** | **99.4** | **91.4** | **89.7** | **86.7** | **99.0** | **97.3** | **97.9** | **94.5** | **88.9** | **88.3** | **63.4** | **70.5** | **48.1** |

The results are summarized in Table 1. From the table, it is evident that SATC consistently outperforms other approaches across all datasets. Specifically, SATC achieves the highest accuracy, attaining an accuracy of 99.8% on ImageNet-10, with improvements over the second-best method (TAC) of 7.0% on ImageNet-Dogs and 4.4% on CIFAR-100, among others. In terms of NMI and ARI, SATC also leads, with substantial gains over competing methods. These results highlight that the SATC effectively improves the clustering performance through integrating spatial structural information and selectively incorporates textual features in a synergistic manner to facilitate image clustering. Moreover, the consistent gains across datasets demonstrate that SATC is robust and generalizable, effectively handling complex intra-class and inter-class variations while enhancing clustering accuracy and stabilizing the learning process.

## 4.3 ABLATION STUDIES

### 4.3.1 ANALYSIS ABOUT TEXTUAL FEATURE SELECTOR

To validate the effectiveness of our compactness-aware textual feature selector, we evaluate clustering performance with and without leveraging the textual modality across 18 benchmark datasets.

As described in Section 3.2.2, the compactness $\tau$ measures the intra-cluster concentration of textual features: lower $\tau$ indicates that textual features are highly clustered and potentially semantically redundant, while higher $\tau$ suggests greater semantic diversity, making the textual modality more informative for clustering.

From Table 2, we observe a clear correlation between the compactness $\tau$ and the usefulness of textual information: datasets with lower $\tau$ values ($\tau < 0.33$) generally show minimal or even negative benefits from incorporating textual features, while datasets with higher $\tau$ values ($\tau > 0.33$) tend to benefit significantly. According to the experimental results, the selection of a textual compactness threshold $0.33$ obtains the optimal results on the 18 datasets. We would like to clarify that $0.33$ could be understood as an empirically discovered rule derived from broad bench-

Table 2: The results of the textual feature selector, including compactness $\tau$, the clustering ACC with and without textual features, and the indicator of the textual feature selector. **Bold** indicates the best ACC under the Use text or No text. Gray Shading denotes the result with the proposed textual feature selector. Dataset indices refer to Appendix A.

| Dataset | $\tau$ | No Text | Use Text | $\tau > 0.33$ |
|---|---|---|---|---|
| 1 | 0.1083 | **89.8** | 60.7 | ✗ |
| 2 | 0.1241 | **85.8** | 59.9 | ✗ |
| 3 | 0.1476 | **99.8** | 99.6 | ✗ |
| 4 | 0.1541 | **36.2** | 34.5 | ✗ |
| 5 | 0.1787 | **76.0** | 74.0 | ✗ |
| 6 | 0.1892 | **31.5** | 26.1 | ✗ |
| 7 | 0.2027 | **86.3** | 83.4 | ✗ |
| 8 | 0.2247 | **91.4** | 87.0 | ✗ |
| 9 | 0.3084 | **81.4** | 72.2 | ✗ |
| 10 | 0.3222 | **89.4** | 84.9 | ✗ |
| 11 | 0.3226 | **52.3** | 49.6 | ✗ |
| 12 | 0.3349 | 93.6 | **94.5** | ✓ |
| 13 | 0.3623 | 60.6 | **67.1** | ✓ |
| 14 | 0.3780 | 98.5 | **99.0** | ✓ |
| 15 | 0.3839 | 52.2 | **65.2** | ✓ |
| 16 | 0.4155 | 56.7 | **63.4** | ✓ |
| 17 | 0.4361 | 51.6 | **60.5** | ✓ |
| 18 | 0.4822 | 47.4 | **52.0** | ✓ |

marking, rather than a hyperparameter tuned on test performance. This value was identified based on aggregated trends observed across all 18 datasets in our study. Importantly, it was not iteratively optimized for any specific dataset's test performance.

The results show that the proposed textual feature selector exactly matches the highest ACC values under both the "Use Text" and "No Text" settings for each dataset, demonstrating that by leveraging the textual compactness $\tau$, SATC can adaptively select textual features that are beneficial for downstream clustering.

### 4.3.2 ANALYSIS OF SPATIAL MODELING ARCHITECTURES

In our framework, spatial feature extraction involves two key components: ResNet-50 and GAT. ResNet-50 first divides the input image into patches and generates node features for each patch, providing rich local spatial representations. The GAT then models relational dependencies among these patches. To evaluate their individual contributions, we conducted ablation studies comparing: (1) using only ResNet-50 features, (2) replacing GAT with GCN or Transformer.

Table 3 presents our ablation study on spatial components. The first column provides baseline results: the upper section shows K-Means applied to raw CLIP feature, while the lower section incorporates textual feature through mutual distillation. In the second column, we introduce the ResNet-50 node feature, with the lower section further including the textual feature. Columns 3–5 compare different spatial modeling choices.

Comparing the first two columns, we observe that ResNet-50 node features improve clustering accuracy over the raw CLIP feature. In addition, results show that spatial modeling consistently improves performance, with GAT achieving the highest accuracy versus GCN and Transformer. GAT's advantage may come from its adaptive attention mechanism, which dynamically weights neighboring patches rather than using uniform aggregation (GCN) or fixed receptive fields (Transformer).

Table 3: Comparison of spatial modeling (ACC%).

| CLIP | ✓ | ✓ | ✓ | ✓ | ✓ |
| ResNet-50 | | ✓ | ✓ | ✓ | ✓ |
| Choice | NONE | NONE | GCN | TRANS | GAT |
|---|---|---|---|---|---|
| 1 | 56.5 | 75.2 | 89.7 | 76.2 | **89.8** |
| 2 | 62.3 | 78.2 | 76.1 | 78.4 | **85.8** |
| 3 | 98.6 | 99.6 | 99.8 | 99.7 | **99.8** |
| 4 | 31.6 | 34.7 | **38.2** | 34.7 | 36.2 |
| 5 | 64.2 | 72.5 | 74.3 | 72.5 | **76.0** |
| 6 | 27.0 | 30.7 | 31.4 | 31.2 | **31.5** |
| 7 | 64.3 | 81.6 | 85.8 | **89.5** | 86.3 |
| 8 | 38.9 | 84.1 | 86.7 | 84.5 | **91.4** |
| 9 | 67.4 | 78.3 | 78.7 | 73.2 | **81.4** |
| 10 | 52.4 | 75.1 | 86.0 | 87.6 | **89.4** |
| 11 | 46.5 | 48.5 | 50.2 | 49.5 | **57.4** |
| 12 | 92.2 | 94.0 | 94.2 | 94.0 | **94.5** |
| 13 | 67.6 | 67.2 | **69.7** | 69.6 | 67.1 |
| 14 | 98.3 | 98.7 | 98.9 | 98.7 | **99.0** |
| 15 | 62.9 | 61.4 | 65.0 | 64.4 | **65.2** |
| 16 | 59.0 | 59.3 | 59.3 | 56.5 | **63.4** |
| 17 | 58.6 | 55.3 | 58.9 | 54.7 | **60.5** |
| 18 | 46.6 | 49.3 | 51.2 | **52.3** | 52.0 |
| **Avg.** | 60.8 | 69.1 | 71.9 | 70.4 | **73.7** |

The detailed analysis of whether performance improvement comes from the method design or from the use of more powerful pre-trained features is given in the Appendix M.

### 4.3.3 ANALYSIS ABOUT THE SPATIAL FEATURE

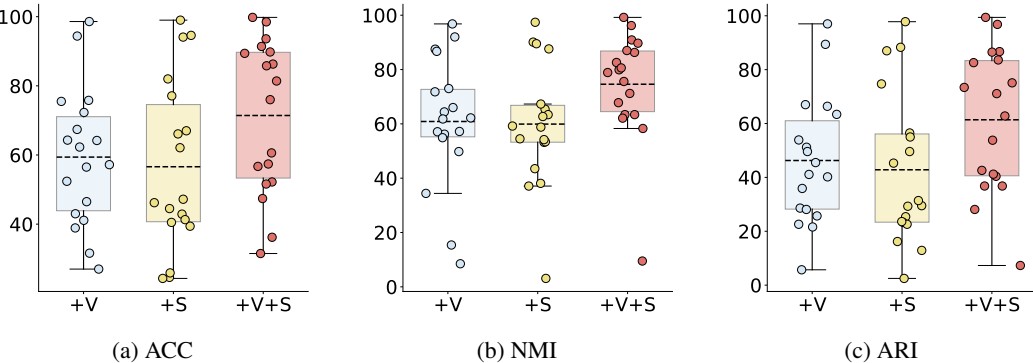

(a) ACC        (b) NMI        (c) ARI

Figure 3: Average clustering performance across 18 datasets using visual features via K-Means (+V), spatial features via K-Means (+S), and their combination via mutual distillation (+V+S).

To verify the effectiveness of spatial features in visual representation, we applied K-Means clustering on two types of features: visual features extracted by CLIP (+V) and spatial features alone (+S). In addition, we performed clustering on a combination of visual and spatial features using mutual distillation (+V+S) as implemented in our SATC framework. Figure 3 shows the average performance across 18 datasets, with the detailed results for each dataset provided in Appendix G.

From Figure 3, it can be observed that clustering on visual features (+V) and spatial features alone (+S) achieves comparable overall performance across the 18 datasets, indicating that both modalities capture valuable but distinct information. When the two are combined through mutual distillation (+V+S), the average clustering results surpass those of using a single modality. This suggests that visual and spatial features provide complementary signals: the spatial modality offers structural cues that enhance the discriminative power of the visual modality, while the visual modality enriches the semantic meaning of the spatial patterns. These results show that combining visual and spatial features improves clustering quality and robustness.

## 4.4 PARAMETER ANALYSIS

To investigate the influence of the balancing hyperparameters $\lambda_1$ and $\lambda_2$ in our training objective (Eq. 6), we conduct parameter analysis experiments on CIFAR-10 and ImageNet-10. Here, $\lambda_1$ controls the weight of the consistency loss and $\lambda_2$ regulates the entropy loss.

Figure 4 shows that SATC maintains stable performance across a wide range of $\lambda_1$ and $\lambda_2$. Especially on ImageNet-10, the model is not only highly robust but also consistently achieves over 99.6% accuracy.

These results indicate that $\lambda_1$ and $\lambda_2$ play complementary roles: $\lambda_1$ encourages agreement across modalities, while $\lambda_2$ prevents overconfident unimodal predictions.

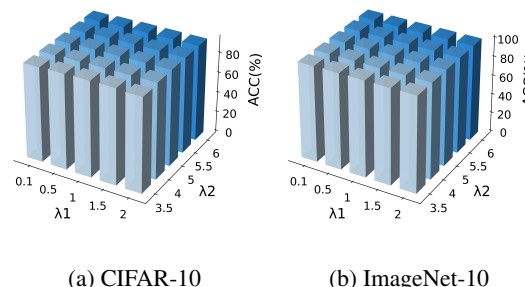

(a) CIFAR-10    (b) ImageNet-10

Figure 4: Parameter analysis of $\lambda_1$ and $\lambda_2$ on CIFAR-10 and ImageNet-10. The 3D bar charts show clustering ACC (%) under different hyperparameter combinations.

## 4.5 TIME ANALYSIS

All experiments are conducted on a single Nvidia RTX 4060 Ti GPU. Figure 5 presents a detailed comparison of running time and accuracy between SATC and TAC across 18 datasets. Overall, SATC achieves higher clustering accuracy and lower computation time compared with TAC on most of the datasets, highlighting the efficiency and effectiveness of SATC. The ACC and running time for each dataset are provided in Appendix C.

## 4.6 TOP-30 DISCRIMINATIVE NOUNS FOR FOUR REPRESENTATIVE DATASETS

In Appendix I, we list the top-30 discriminative nouns and word cloud visualizations for two datasets where textual features are utilized (CIFAR-10, STL-10) and two where they are excluded (MNIST, FER2013). The nouns show that for datasets where textual features are harmful, the nouns are largely unrelated to the

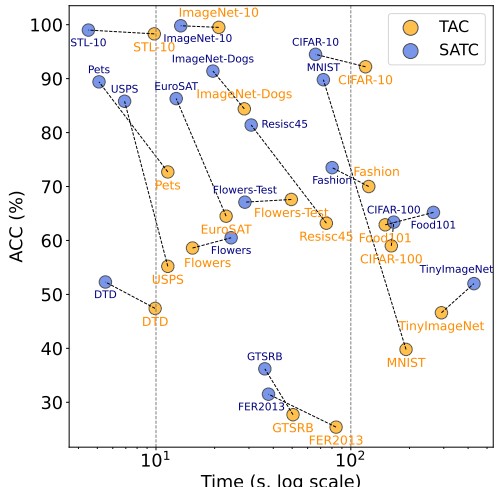

Figure 5: Comparison of ACC and Times for SATC and TAC across 18 datasets.

visual content. In contrast, datasets where text is beneficial exhibit semantically relevant terms. The word cloud visualizations clearly demonstrate that higher $\tau$ datasets exhibit richer lexical diversity and semantic relevance, while lower $\tau$ datasets display limited.

## 5 CONCLUSION

In this work, we propose an image clustering approach that utilizes spatial structure and selective text to jointly facilitate image clustering (SATC). SATC employs GAT to effectively model spatial relationships among image patches while adaptively integrating textual features through a novel textual feature selector. The framework further enhances performance through a tri-modal mutual distillation framework that optimally fuses visual, spatial, and textual modalities. Extensive experimental results demonstrate the effectiveness of the proposed SATC and the rationality of the textual feature selector. Future research directions include integrating more diverse prior knowledge and developing advanced fusion strategies to enhance performance.

ACKNOWLEDGMENTS

This work was supported by the National Natural Science Foundation of China (Nos. 62476160, 62441239, T2495251, 62136005, 62306170, U24A20253), the Special Fund for Science and Technology Innovation Teams of Shanxi Province (No. 202304051001001).

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

## APPENDIX

The appendix is organized as follows:

- A Datasets
- B Clustering Performance Comparison
- C Running Time Results
- D Visualization Analysis
- E Loss Function Analysis
- F Distillation Direction Analysis
- G Spatial Feature Analysis
- H SATC Algorithm Framework
- I Top-30 Discriminative Nouns for Four Representative Datasets
- J Comparison of Visual and Visual+Textual Clustering Performance
- K Comparison of Different Modality Cluster Heads
- L Theoretical Guidance for Utilizing Textual Compactness
- M The Contribution of ResNet-50 to Model Performance

## A DATASETS

We evaluate our framework on 18 widely used benchmark datasets, covering a diverse range of vision tasks. These include general object classification datasets CIFAR-10 (DeVries & Taylor, 2017), CIFAR-100 (DeVries & Taylor, 2017), STL-10 (Coates et al., 2011), TinyImageNet (Le & Yang, 2015), and ImageNet-10 (Chang et al., 2017); fine-grained object classification datasets Food101 (Bossard et al., 2014), Flowers (Nilsback & Zisserman, 2008), Flowers(Test) (Nilsback & Zisserman, 2008), ImageNet-Dogs (Chang et al., 2017), and OxfordPets (Parkhi et al., 2012); handwritten digit datasets MNIST (Schott et al., 2018) and USPS (Sankaranarayanan et al., 2018); the fashion dataset Fashion-MNIST (Xiao et al., 2017); the texture dataset DTD (Cimpoi et al., 2014); the facial emotion recognition dataset FER2013 (Goodfellow et al., 2013); the satellite image classification datasets EuroSAT (Helber et al., 2019) and RESISC45 (Cheng et al., 2017); and the German Traffic Sign Recognition Benchmark (GTSRB) (Stallkamp et al., 2012). The detailed statistics of each dataset are summarized in Table 4.

Table 4: Summary statistics of datasets, including number of classes, train size, and test size.

|  | Dataset | Num Classes | Train Size | Test Size |
|---|---|---|---|---|
| 1 | MNIST (Schott et al., 2018) | 10 | 60,000 | 10,000 |
| 2 | USPS (Sankaranarayanan et al., 2018) | 10 | 7,291 | 2,007 |
| 3 | ImageNet-10 (Chang et al., 2017) | 10 | 10,500 | 2,630 |
| 4 | GTSRB (Stallkamp et al., 2012) | 43 | 26,640 | 12,630 |
| 5 | Fashion (Xiao et al., 2017) | 10 | 60,000 | 10,000 |
| 6 | FER2013 (Goodfellow et al., 2013) | 7 | 28,709 | 7,178 |
| 7 | EuroSAT (Helber et al., 2019) | 10 | 10,000 | 5,000 |
| 8 | ImageNet-Dogs (Chang et al., 2017) | 15 | 15,600 | 3,900 |
| 9 | Resisc45 (Cheng et al., 2017) | 45 | 25,200 | 6,300 |
| 10 | OxfordPets (Parkhi et al., 2012) | 37 | 3,680 | 3,669 |
| 11 | DTD (Cimpoi et al., 2014) | 47 | 3,760 | 1,880 |
| 12 | CIFAR-10 (DeVries & Taylor, 2017) | 10 | 50,000 | 10,000 |
| 13 | Flowers(Test) (Nilsback & Zisserman, 2008) | 102 | 6,149 | 2,040 |
| 14 | STL-10 (Coates et al., 2011) | 10 | 5,000 | 8,000 |
| 15 | Food101 (Bossard et al., 2014) | 101 | 75,750 | 25,250 |
| 16 | CIFAR-100 (DeVries & Taylor, 2017) | 100 | 50,000 | 10,000 |
| 17 | Flowers (Nilsback & Zisserman, 2008) | 102 | 2,040 | 6,149 |
| 18 | TinyImageNet (Le & Yang, 2015) | 200 | 100,000 | 10,000 |

# B Clustering Performance Comparison

Table 5 presents a comprehensive comparison of clustering performance for ClipKmeans, TAC, and our proposed SATC across 18 widely used benchmark datasets, measured in terms of ACC (%), NMI (%), and ARI (%). Overall, SATC consistently achieves the highest performance on the majority of datasets, with average ACC, NMI, and ARI of 73.7%, 76.3%, and 63.8%, respectively. This demonstrates that SATC effectively exploits both rich spatial features and selectively incorporated textual information, producing clusters that are not only more accurate but also more robust and semantically meaningful, compared with the visual-only ClipKmeans baseline (59.4% / 60.9% / 46.3%) and the text-enhanced TAC method (63.1% / 65.6% / 51.3%).

The performance gains of SATC are particularly prominent on datasets with complex visual structures or high intra-class variability. For instance, on MNIST, ACC increases dramatically from 56.5% (ClipKmeans) and 39.8% (TAC) to 89.8% with SATC, while EuroSAT, CIFAR-100, and GT-SRB also exhibit substantial improvements across all three metrics. These results indicate that the integration of selective textual cues with spatial and visual representations allows SATC to better capture underlying class distributions, thereby improving clustering quality in challenging scenarios where purely visual features may fall short.

Although a few datasets, such as Flowers(Test) and Flowers, show marginally higher ACC or NMI for Clip, SATC still demonstrates superior overall performance across nearly all datasets and metrics. The aggregated average values reported in the last row as AVG further confirm the effectiveness and generalizability of SATC. Taken together, these results validate that SATC not only enhances clustering accuracy but also produces clusters that are more consistent with semantic labels, highlighting its potential as a robust and versatile approach for large-scale image clustering tasks.

Table 5: Clustering performance (ACC% / NMI% / ARI%) on benchmark datasets.

| Dataset | ACC% | | | NMI% | | | ARI% | | |
|---|---|---|---|---|---|---|---|---|---|
| | Clip | TAC | SATC | Clip | TAC | SATC | Clip | TAC | SATC |
| MNIST | 56.5 | 39.8 | **89.8** | 54.9 | 31.5 | **87.0** | 35.9 | 22.3 | **83.6** |
| USPS | 62.3 | 55.2 | **85.8** | 61.8 | 51.0 | **80.6** | 51.2 | 44.0 | **73.4** |
| ImageNet-10 | 98.6 | 99.5 | **99.8** | 96.8 | 98.5 | **99.2** | 97.0 | 98.8 | **99.4** |
| GTSRB | 31.6 | 27.7 | **36.2** | 49.8 | 43.7 | **58.3** | 21.6 | 18.9 | **28.1** |
| Fashion | 64.2 | 70.0 | **76.0** | 62.2 | 66.2 | **71.2** | 49.6 | 56.7 | **62.8** |
| FER2013 | 27.0 | 25.4 | **31.5** | 8.5 | 6.0 | **9.5** | 5.7 | 4.4 | **7.3** |
| EuroSAT | 64.3 | 64.5 | **86.3** | 15.4 | 53.3 | **78.9** | 45.5 | 50.0 | **75.1** |
| ImageNet-Dogs | 38.9 | 84.4 | **91.4** | 34.4 | 77.4 | **89.7** | 22.6 | 72.0 | **86.7** |
| Resisc45 | 67.4 | 63.2 | **81.4** | 73.0 | 70.9 | **82.6** | 53.9 | 50.0 | **71.1** |
| OxfordPets | 52.4 | 72.7 | **89.4** | 66.0 | 79.1 | **90.9** | 40.2 | 61.2 | **82.6** |
| DTD | 46.5 | 47.4 | **57.4** | 56.2 | 57.3 | **63.4** | 28.6 | 32.1 | **40.4** |
| CIFAR-10 | 75.8 | 92.2 | **94.5** | 71.8 | 83.7 | **88.9** | 63.4 | 83.6 | **88.3** |
| Flowers(Test) | **72.3** | 67.6 | 67.1 | **86.7** | 84.3 | 85.9 | **67.0** | 63.4 | 63.2 |
| STL-10 | 94.4 | 98.3 | **99.0** | 92.0 | 95.7 | **97.3** | 89.4 | 96.3 | **97.9** |
| Food101 | 57.2 | 62.9 | **65.2** | 64.4 | 69.0 | **71.1** | 41.1 | 47.4 | **50.7** |
| CIFAR-100 | 43.0 | 59.0 | **63.4** | 57.1 | 68.6 | **70.5** | 28.1 | 42.7 | **48.1** |
| Flowers | **75.5** | 58.6 | 60.5 | **87.5** | 83.2 | 83.0 | **66.4** | 54.1 | 53.8 |
| TinyImageNet | 41.1 | 46.6 | **52.0** | 57.2 | 61.4 | **65.1** | 25.7 | 31.6 | **36.8** |
| **AVG.** | 59.4 | 63.1 | **73.7** | 60.9 | 65.6 | **76.3** | 46.3 | 51.3 | **63.8** |

# C  TIME ANALYSIS

In this section, we present the running time and clustering accuracy comparisons between SATC and the TAC baseline across 18 benchmark datasets. Figure 6 illustrates the results.

It can be observed that SATC generally achieves higher ACC while maintaining competitive or even lower running times compared to TAC, demonstrating both efficiency and effectiveness. In particular, SATC shows clear advantages on several challenging datasets such as CIFAR-100, Food101, and TinyImageNet, where the improvements in clustering accuracy are substantial. Moreover, on grayscale datasets such as USPS and MNIST, SATC also achieves remarkably high ACC while offering shorter running times compared to TAC. These observations suggest that SATC is not only effective in boosting clustering performance but also scalable to datasets of different sizes and complexity levels.

Overall, these results demonstrate that SATC consistently provides higher clustering performance while remaining time-efficient across diverse datasets, highlighting its practical applicability for large-scale image clustering tasks.

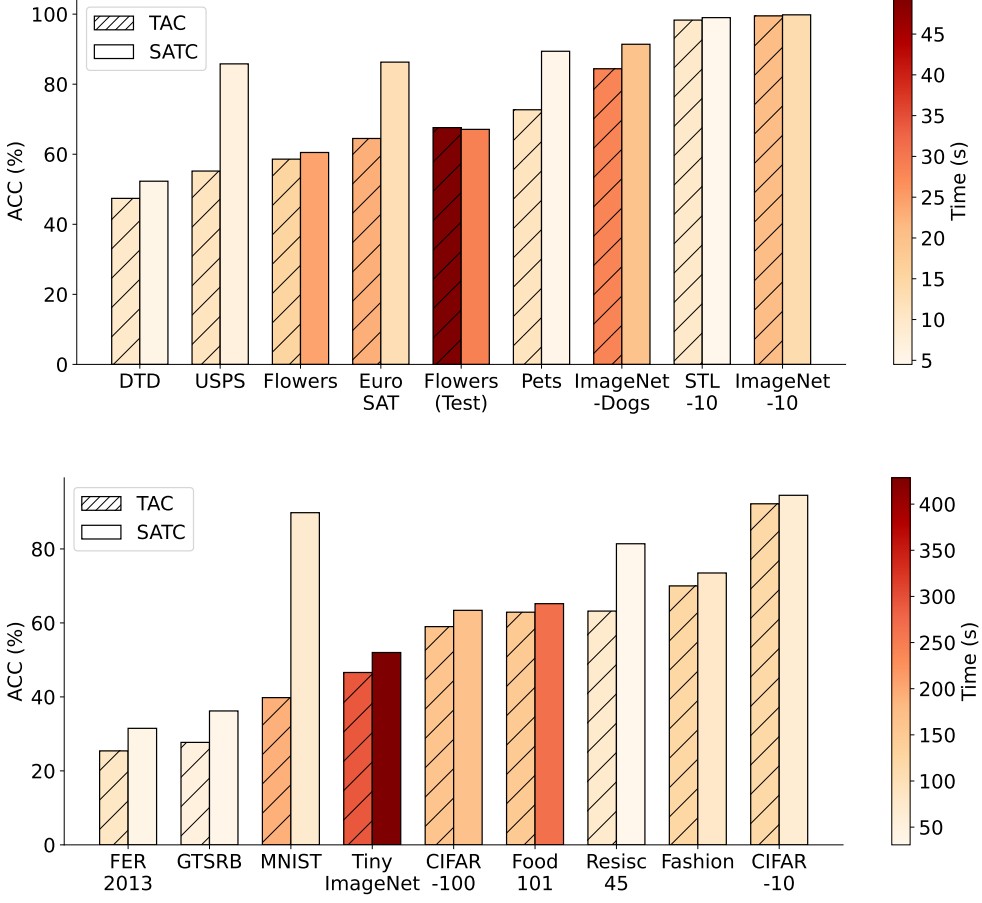

Figure 6: Clustering accuracy (ACC%) and running time comparison across all 18 datasets. For each dataset, the bars represent clustering accuracy (ACC%) for SATC and TAC, while the color intensity corresponds to the running time in seconds.

## D  VISUALIZATION ANALYSIS

To better understand the impact of each modality on clustering performance, we conducted t-SNE visualizations on OxfordPets and CIFAR-10 datasets. First, we define five different modality configurations for analysis:

- ClipKmeans: This method applies K-Means clustering on the raw visual features extracted by CLIP.
- W/O Visual: This variant omits the visual modality.
- W/O Spatial: This variant omits the spatial modality.
- W/O Textual: This variant omits the textual modality.
- Tri-Modal: This variant includes all three modalities.

The t-SNE plots in Figure 7 provide an intuitive illustration of clustering behavior under different configurations.

Visual features are essential for clustering performance. As shown in the W/O Visual configuration, without visual features, relying solely on spatial and textual information fails to produce meaningful cluster assignments, and the data becomes completely inseparable. This clearly demonstrates that visual features are indispensable, while spatial and textual features serve only as auxiliary cues to enhance clustering performance. Spatial structure significantly enhances discriminability: comparing W/O Spatial with Tri-Modal shows that including spatial feature helps form more compact and well-separated clusters, especially for datasets like OxfordPets, where fine-grained object details and relative positions are important. Textual information plays a complementary role in refining clustering results. By comparing W/O Textual with Tri-Modal on the CIFAR-10 dataset, we observe that beneficial textual information can further enhance image clustering. However, treating all textual information as equally useful without selective adaptation may harm clustering performance, as shown on the OxfordPets dataset.

SATC model (highlighted with red bounding boxes) achieving the highest ACC values on both datasets. SATC first demonstrates the ability to accurately discard textual information that may harm clustering performance. Furthermore, by combining visual, spatial, and beneficial textual modalities, the model can achieve better clustering performance. These visualizations qualitatively support our ablation study: each modality contributes uniquely to clustering performance.

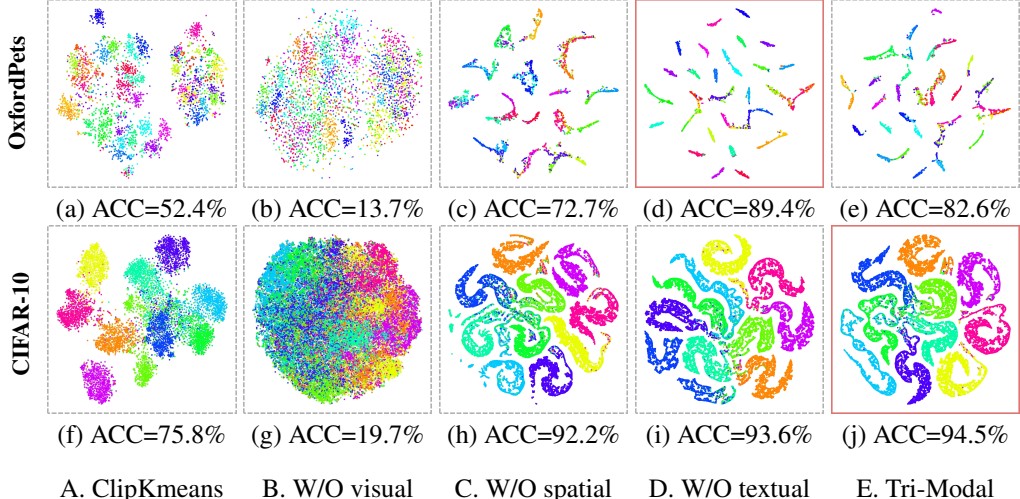

Figure 7: t-SNE visualizations of clustering results on the OxfordPets (top) and CIFAR-10 (bottom) under different modality configurations. Red bounding boxes indicate the performance of our proposed method SATC.

# E  LOSS FUNCTION ANALYSIS

We investigate how individual loss functions ($\mathcal{L}_{\text{distill}}$, $\mathcal{L}_{\text{consist}}$, and $\mathcal{L}_{\text{entropy}}$) and their combinations affect clustering performance across six benchmark datasets. These losses are designed to align cluster assignment distributions across modalities, enforce consistency, and encourage confident assignments, as described in Section 3.3. Their respective contributions and interactions are summarized in Figure 8.

When applied individually, the three losses influence the clustering in complementary ways: $\mathcal{L}_{\text{distill}}$, which aligns the cluster distributions between modalities and plays a dominant role in transferring spatial structure knowledge and textual information. $\mathcal{L}_{\text{consist}}$ acts as a regularizer, enforcing local consistency across modalities and stabilizing training; while it achieves lower standalone ACC, it improves normalized mutual information on datasets with high intra-class variability. $\mathcal{L}_{\text{entropy}}$ prevents degenerate solutions by promoting confident and well-separated cluster assignments, but is insufficient alone to capture cross-modal information. Combining pairs of loss functions improves ACC compared to using each loss individually. For example, the combination of $\mathcal{L}_{\text{distill}}$ and $\mathcal{L}_{\text{entropy}}$ increases ACC on MNIST to 80.0%, indicating that multi-modal knowledge alignment and entropy maximization work synergistically. Similarly, the pairs of $\mathcal{L}_{\text{consist}}$ with

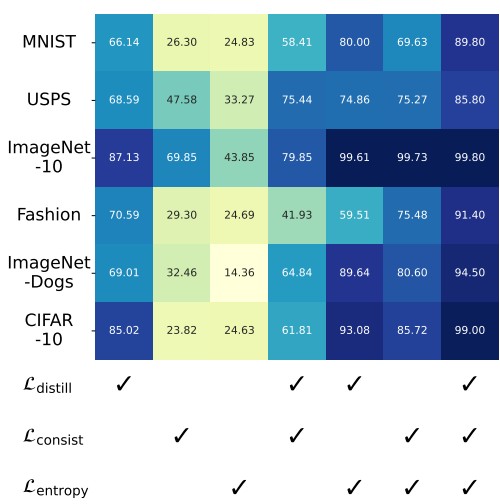

Figure 8: ACC with different loss combinations on 6 benchmark datasets.

$\mathcal{L}_{\text{entropy}}$ outperform their respective individual losses, demonstrating that entropy maximization effectively complements consistency. Integrating all three loss functions ($\mathcal{L}_{\text{distill}} + \mathcal{L}_{\text{consist}} + \mathcal{L}_{\text{entropy}}$) achieves the best ACC across all evaluated datasets.

From Figure 9, the NMI and ARI metrics exhibit trends consistent with ACC, showing substantial improvements in clustering quality. Taken together, these results indicate that the three losses—distillation, consistency, and entropy—interact synergistically, indicating that these three losses interact positively and complement each other.

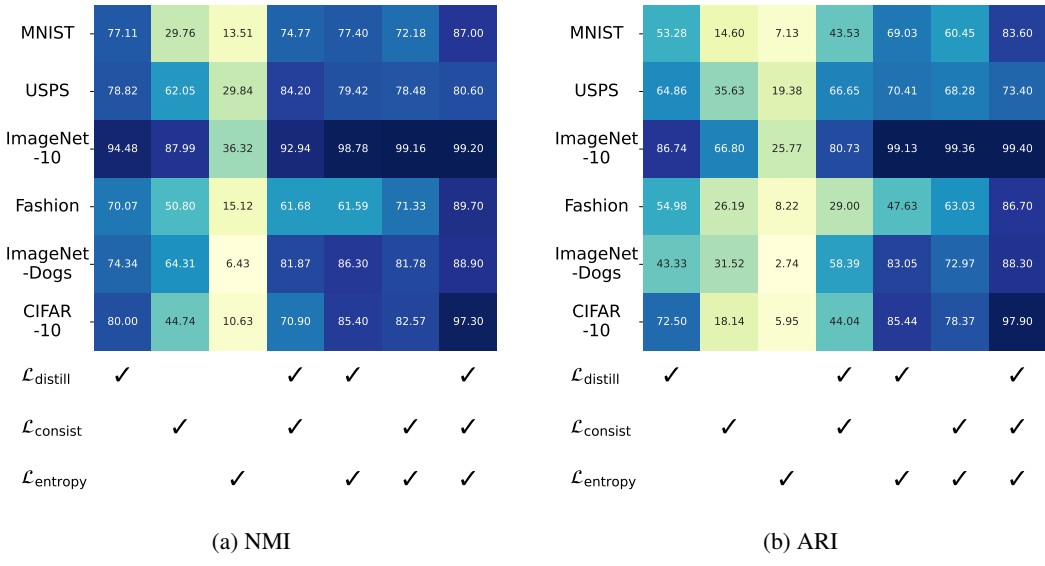

(a) NMI  (b) ARI

Figure 9: NMI and ARI with different loss combinations on 6 benchmark datasets.

# F    DISTILLATION DIRECTION ANALYSIS

Table 6 reports the clustering performance of SATC under different distillation directions across 18 benchmark datasets. S→V denotes distillation from the spatial modality to the visual modality, V→S represents distillation from the visual modality to the spatial modality, and S↔V corresponds to mutual distillation between the two.

Overall, mutual distillation consistently achieves the best results, yielding the highest average scores of 73.7% ACC, 76.3% NMI, and 63.8% ARI, and outperforming either single-direction alternative on most datasets such as MNIST, USPS, Fashion, and CIFAR-100. This superiority can be attributed to the fact that spatial and visual representations encode complementary forms of structural and semantic information. The bidirectional exchange of supervisory signals enables both modalities to refine their feature spaces in a coordinated manner, leading to clusters that are not only more compact but also better aligned with semantic categories.

When restricted to single-direction distillation, the performance becomes more dataset-dependent. S→V often provides advantages on datasets such as CIFAR-10, Flowers, and TinyImageNet, where the spatial clustering head captures low-level structural cues that help the visual modality avoid collapsing into overly coarse groupings. In contrast, V→S achieves stronger results on more complex and visually diverse datasets such as ImageNet-Dogs, EuroSAT, and OxfordPets, where the visual modality offers richer semantic guidance that stabilizes the spatial clustering head. Despite these complementary strengths, both single-direction strategies lack the reciprocal feedback that allows each modality to iteratively correct the other's biases.

The general trend across ACC, NMI, and ARI is therefore consistent: leveraging bidirectional distillation between spatial and visual modalities provides the most robust clustering performance. By jointly exploiting the semantic richness of visual features and the structural discrimination of spatial features, mutual distillation enables SATC to learn more discriminative and semantically consistent clusters, especially in challenging datasets where single-direction supervision proves insufficient.

Table 6: The clustering performance (ACC% / NMI% / ARI%) of SATC with different distillation directions on 18 benchmark datasets. S:Use the spatial clustering head to generate the final cluster assignments.

| Dataset | ACC% | | | NMI% | | | ARI% | | |
|---|---|---|---|---|---|---|---|---|---|
| | S→V | V→S | S↔V | S→V | V→S | S↔V | S→V | V→S | S↔V |
| MNIST | 78.8 | 88.4 | **89.8** | 77.9 | 85.2 | **87.0** | 68.3 | 81.7 | **83.6** |
| USPS | 78.4 | 79.7 | **85.8** | **80.9** | 77.8 | 80.6 | **73.6** | 68.2 | 73.4 |
| ImageNet-10 | 99.8 | 99.7 | **99.8** | **99.3** | 99.2 | 99.2 | **99.5** | 99.4 | 99.4 |
| GTSRB | **37.0** | 30.6 | 36.2 | **58.6** | 53.4 | 58.3 | **29.3** | 24.5 | 28.1 |
| Fashion | 72.0 | 68.8 | **76.0** | 70.9 | 66.5 | **71.2** | 61.5 | 56.4 | **62.8** |
| Fre2013 | 29.2 | 28.7 | **31.5** | 8.6 | 7.5 | **9.5** | 6.1 | 5.6 | **7.3** |
| EuroSAT | 80.8 | **88.7** | 86.3 | 76.2 | **81.0** | 78.9 | 71.1 | **78.4** | 75.1 |
| ImageNet-Dogs | 83.3 | **91.8** | 91.4 | 82.4 | **90.4** | 89.7 | 74.7 | **87.5** | 86.7 |
| Resisc45 | 79.7 | 71.6 | **81.4** | 81.7 | 77.0 | **82.6** | 73.6 | 60.3 | 71.1 |
| OxfordPets | 80.9 | 87.7 | **89.4** | 86.2 | 90.7 | **90.9** | 72.3 | 82.1 | **82.6** |
| DTD | 52.5 | 52.1 | **57.4** | 62.0 | 61.5 | **63.4** | 37.2 | 37.0 | **40.4** |
| CIFAR-10 | **95.0** | 93.8 | 94.5 | 88.7 | 86.6 | **88.9** | **89.3** | 87.0 | 88.3 |
| Flowers(test) | 68.3 | 66.6 | **67.1** | 85.9 | 85.4 | **85.9** | 63.4 | 61.7 | 63.2 |
| STL-10 | 98.9 | 98.8 | **99.0** | 97.1 | 96.9 | **97.3** | 97.5 | 97.3 | **97.9** |
| food101 | 65.2 | 63.6 | **65.2** | 69.7 | 70.4 | **71.1** | 49.8 | 49.1 | **50.7** |
| CIFAR-100 | 59.0 | 56.4 | **63.4** | 69.0 | 67.3 | **70.5** | 45.0 | 42.5 | **48.1** |
| Flowers | **73.2** | 72.3 | 60.5 | **87.2** | 87.0 | 83.0 | **64.7** | 64.4 | 53.8 |
| TinyImageNet | **53.5** | 51.8 | 52.0 | **65.9** | 64.9 | 65.1 | **38.4** | 36.9 | 36.8 |
| **AVG.** | 71.4 | 71.7 | **73.7** | 74.9 | 74.9 | **76.3** | 62.0 | 62.2 | **63.8** |

# G    SPATIAL FEATURE ANALYSIS

To better understand the role of spatial feature in image clustering, we present the detailed clustering performance on each dataset. From Figure 10, it can be observed that clustering using visual features (+V) and spatial features alone (+S) achieves comparable overall performance across the 18 datasets, indicating that both modalities capture valuable but distinct information. Considering the seven representative datasets, they can be roughly grouped into three categories: handwritten digits (MNIST, USPS), natural object images (OxfordPets, CIFAR-10, TinyImageNet), and aerial/remote-sensing images (EuroSAT, Resisc45). For the handwritten digits, ACC is 56.5% (+V) vs. 41.3% (+S) on MNIST and 62.3% vs. 67.0% on USPS, showing that spatial features already capture much of the structural information. For natural object images, ACC is 52.4% vs. 82.0% on OxfordPets, 75.8% vs. 77.1% on CIFAR-10, and 41.1% vs. 40.5% on TinyImageNet, indicating that visual features are generally stronger but spatial features remain complementary. For aerial/remote-sensing images, ACC is 64.3% vs. 66.1% on EuroSAT and 67.4% vs. 47.2% on Resisc45, suggesting that both modalities contribute meaningful structural and semantic cues.

When the two modalities are combined through mutual distillation (+V+S), the clustering performance consistently surpasses that of using a single modality. For instance, ACC increases to 89.8% (+V+S) on MNIST, 85.8% on USPS, 89.4% on OxfordPets, 93.6% on CIFAR-10, 47.4% on TinyImageNet, 86.3% on EuroSAT, and 81.4% on Resisc45. This demonstrates that visual and spatial features provide complementary signals: the spatial modality enhances structural discrimination, while the visual modality enriches semantic understanding of spatial patterns.

Overall, these results empirically demonstrate that integrating visual and spatial modalities through mutual distillation significantly improves clustering quality across diverse datasets. The combined representation not only achieves higher accuracy but also enhances the robustness of the image modality in downstream clustering tasks, highlighting the benefits of leveraging both visual and spatial information.

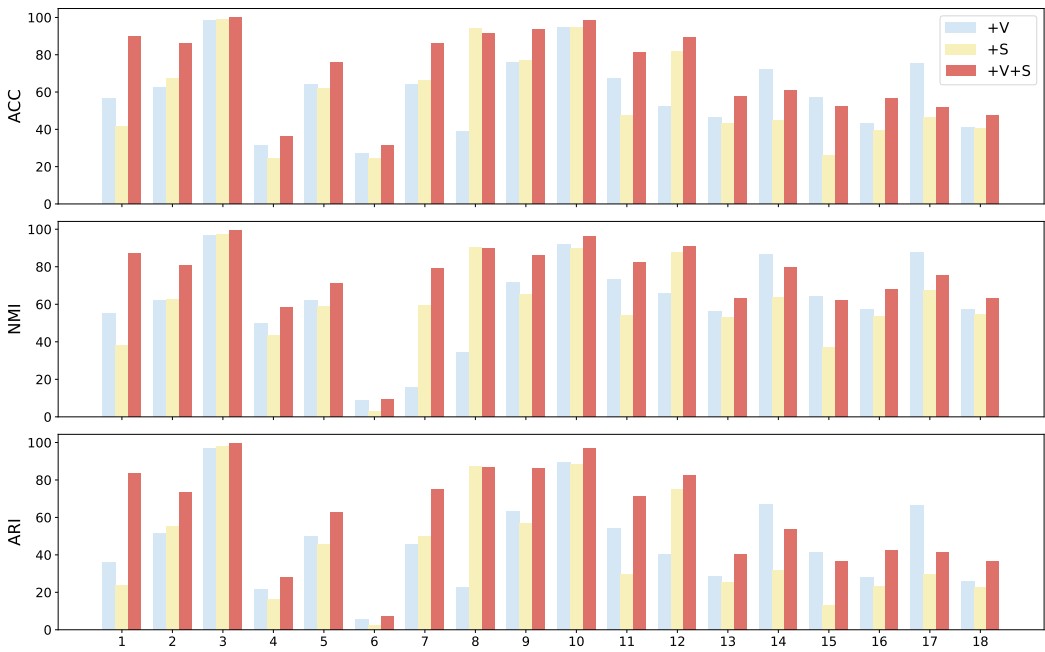

Figure 10: Comparison of clustering performance using three settings: raw visual features with K-Means (+V), spatial features with K-Means(+S), and a combination of visual and spatial features with mutual distillation (+V+S).

# H  SATC Algorithm Framework

---

**Algorithm 1** SATC for Image Clustering

---

**Notations:**

- $N$: number of samples
- $d$: feature dimension predefined by CLIP
- $K$: number of classes in the object dataset
- $\tau$: Textual compactness
- $\overline{\mathcal{L}}_{\text{total}}$: average total loss of the current epoch
- $\mathcal{L}_{\text{best}}$: best average total loss
- $p$: patience counter for early stopping

**Input**: Target raw image dataset $\mathcal{D} = \{x_n\}_{n=1}^N$.

**Process**:

1: Extract visual features $z^{\text{visual}}$ from $\mathcal{D}$ using CLIP.
2: Extract high-level features from $\mathcal{D}$ using a pretrained ResNet-50.
3: Flatten high-level features to form node features $X \in \mathbb{R}^{N \times d}$.
4: Feed $X$ and edge set $E$ into GAT to obtain updated node features $X'$.
5: Apply global average pooling over $X'$ to get spatial embedding $z^{\text{spatial}} \in \mathbb{R}^d$.
6: Extract textual features $z^{\text{textual}}$ from $\mathcal{D}$ using Eq. 3.
7: Compute textual compactness from $z^{\text{textual}}$ using Eq. 5.
8: Freeze the pre-trained features($z^{\text{visual}}, z^{\text{spatial}}, z^{\text{textual}}$).
9: **if** $\tau > 0.33$(Computed in Table 2) **then**
10:     Enable the textual feature $z^{\text{textual}}$.
11: **else**
12:     Do not use $z^{\text{textual}}$.
13: **end if**
14: Initialize $\mathcal{L}_{\text{best}} \leftarrow +\infty$ and patience counter $p \leftarrow 0$.
15: **for** epoch $= 1$ to $200$ **do**
16:     Project $z^{\text{visual}}, z^{\text{spatial}}$ (and $z^{\text{textual}}$ if enabled) into cluster assignment distributions $c^{\text{visual}}, c^{\text{spatial}}, c^{\text{textual}}$ via MLP cluster heads.
17:     Compute total loss $\mathcal{L}$ using Eq. 6.
18:     Update cluster heads via backpropagation.
19:     Compute the average total loss $\overline{\mathcal{L}}_{\text{total}}$ over the current epoch.
20:     **if** $\overline{\mathcal{L}}_{\text{total}} < \mathcal{L}_{\text{best}}$ **then**
21:         Update $\mathcal{L}_{\text{best}} \leftarrow \overline{\mathcal{L}}_{\text{total}}$ and reset $p \leftarrow 0$.
22:     **else**
23:         Increment $p \leftarrow p + 1$.
24:     **end if**
25:     **if** $p \geq 10$ **then**
26:         **Break**
27:     **end if**
28: **end for**
29: **Output**: Final visual cluster head cluster assignments $C$

---

## I   TOP-30 DISCRIMINATIVE NOUNS FOR FOUR REPRESENTATIVE DATASETS

To better understand why textual feature help or hurt clustering performance across datasets, we analyze the discriminative nouns extracted from the textual feature. Specifically, for two datasets where textual features are utilized (CIFAR-10, STL-10) and two where they are excluded (MNIST, FER2013), we compute image–noun softmax retrieval weights, rank nouns by their aggregated weights after deduplication, and take the top-30 nouns to form the discriminative noun set and corresponding word cloud.

Table 7 presents the top-30 discriminative nouns selected for different datasets. For datasets where text was beneficial, such as CIFAR-10 (a dataset of natural object images) and STL-10 (a dataset of natural object images designed for unsupervised learning), the selected nouns correspond closely to actual object categories, indicating high textual discriminability. The word cloud visualizations in Figure 11 further illustrate these patterns. In CIFAR-10 and STL-10, the nouns align well with object categories, and the word clouds are dense, reflecting high semantic diversity and strong textual discriminability. In contrast, for datasets where text was harmful, such as MNIST (handwritten digit images) and FER2013 (facial expression images), the nouns are mostly unrelated to the visual content, and the corresponding word clouds are sparse, indicating poor textual diversity and weak textual discriminability.

Notably, the four word clouds are arranged in order of increasing $\tau$ value; higher $\tau$ corresponds to more diverse and complementary textual clusters, which is visually reflected in the richer, denser word clouds of CIFAR-10 and STL-10. This supports our motivation: datasets with higher $\tau$ tend to have textual feature that contribute more meaningfully to clustering performance.

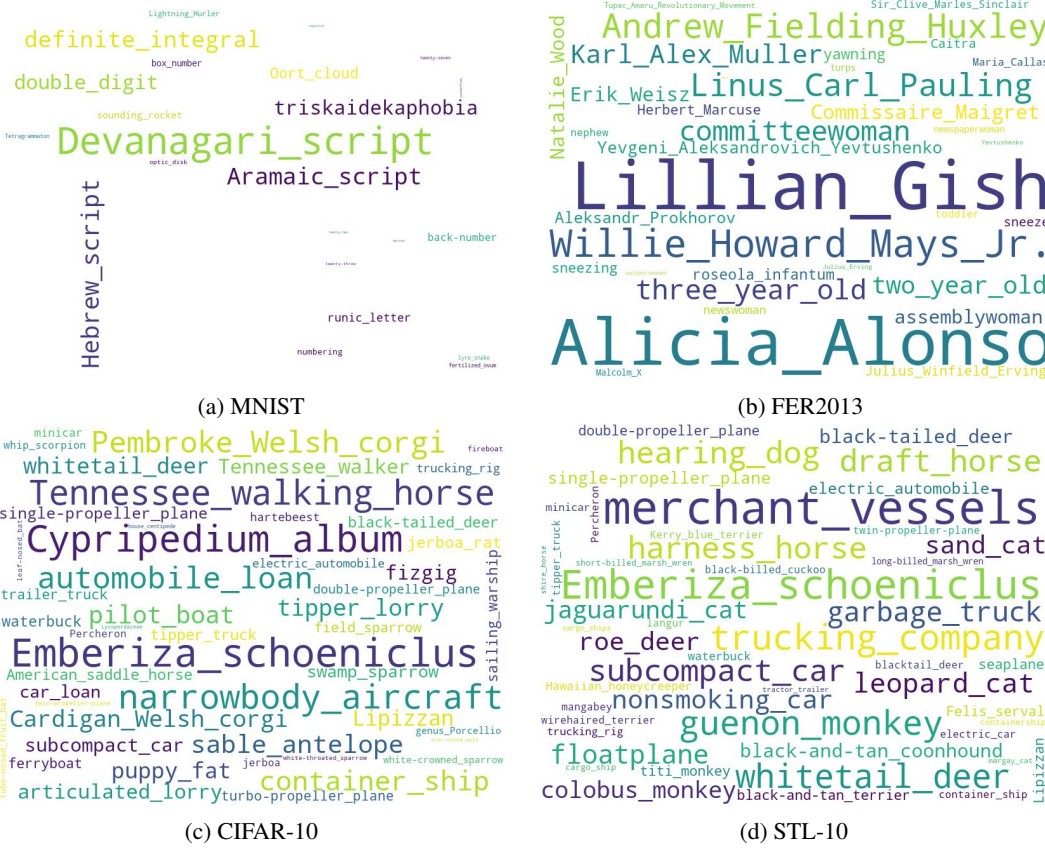

(a) MNIST

(b) FER2013

(c) CIFAR-10

(d) STL-10

Figure 11: Word cloud visualizations of discriminative nouns for four datasets. The size of each word corresponds to its occurrence frequency in the textual features.

Table 7: Top-30 selected discriminative nouns for different datasets, categorized by the effect of text information.

| Text Effect | Dataset | Selected Discriminative Nouns |
|---|---|---|
| Beneficial | CIFAR-10 | Emberiza schoeniclus, Cypripedium album, Tennessee walking horse, narrowbody aircraft, automobile loan, Pembroke Welsh corgi, container ship, sable antelope, pilot boat, Cardigan Welsh corgi, tipper lorry, whitetail deer, puppy fat, Lipizzan, articulated lorry, Tennessee walker, fizgig, subcompact car, swamp sparrow, car loan, single-propeller plane, jerboa rat, American saddle horse, tipper truck, black-tailed deer, waterbuck, sailing warship, trailer truck, ferryboat, field sparrow |
| Beneficial | STL-10 | merchant vessels, Emberiza schoeniclus, trucking company, guenon monkey, whitetail deer, harness horse, hearing dog, draft horse, subcompact car, garbage truck, leopard cat, floatplane, nonsmoking car, sand cat, roe deer, jaguarundi cat, colobus monkey, black-and-tan coonhound, black-tailed deer, single-propeller plane, electric automobile, double-propeller plane, black-and-tan terrier, seaplane, Felis serval, Lipizzan, titi monkey, Hawaiian honeycreeper, waterbuck, trucking rig |
| Harmful | MNIST | Devanagari script, Hebrew script, Aramaic script, definite integral, double digit, triskaidekaphobia, Oort cloud, runic letter, back-number, sounding rocket, box number, numbering, Lightning Hurler, optic disk, fertilized ovum, lyre snake, Tetragrammaton, twenty-seven, twenty-three, cornetfish, equation, twenty-two, operculum, Edward Lear, absorption spectrum, endodontia, needlefish, Hawaiian honeycreeper, retinoblastoma, pied-billed grebe, diamine |
| Harmful | FER2013 | Lillian Gish, Alicia Alonso, Willie Howard Mays Jr., Andrew Fielding Huxley, Linus Carl Pauling, committeewoman, three-year-old, Karl Alex Muller, two-year-old, Commissaire Maigret, Natalie Wood, Erik Weisz, assemblywoman, Yevgeni Aleksandrovich Yevtushenko, Aleksandr Prokhorov, yawning, roseola infantum, sneezing, Herbert Marcuse, Julius Winfield Erving, sneeze, Caitra, newswoman, Maria Callas, nephew, toddler, Sir Clive Marles Sinclair, Malcolm X, newspaperwoman, turps, Tupac Amaru Revolutionary Movement |

## J  COMPARISON OF VISUAL AND VISUAL+TEXTUAL CLUSTERING PERFORMANCE

To investigate how textual features affect clustering, we compare K-Means performance on visual features alone (+V) versus visual features distilled with textual features (+V+T). Visual features are extracted using CLIP ViT-B/32, and textual features are obtained following TAC (Li et al., 2023). This setup enables us to evaluate when textual information enhances or hinders clustering.

Table 8 provides a detailed comparison of clustering performance using only visual features (+V) versus combining visual and textual features (+V+T). The results are divided into two groups: datasets where textual features degrade performance (top section) and datasets where they provide significant improvements (bottom section).

For datasets like MNIST, USPS, GTSRB, FER2013, and Flowers, incorporating textual features actually reduces clustering accuracy, indicating that the textual descriptions either lack discriminative power or introduce semantic noise that misguides the clustering process. Conversely, for datasets such as ImageNet-Dogs, CIFAR-10, STL-10, and OxfordPets, the addition of textual feature leads to substantial performance gains, demonstrating the value of multimodal integration when textual features are semantically meaningful and well-aligned with visual content.

Overall, while the average performance across all 18 datasets shows a modest improvement with the textual feature (59.4% to 63.1%), the dataset-specific variations highlight the importance of selectively incorporating textual guidance based on dataset characteristics.

Table 8: Clustering accuracy (ACC %) with visual (+V) and visual+textual (+V+T) features.

| Dataset | +V | +V+T |
|---|---|---|
| MNIST | **56.5** | 39.8 |
| USPS | **62.3** | 55.2 |
| GTSRB | **31.6** | 27.7 |
| FER2013 | **27.0** | 25.4 |
| Resisc45 | **67.4** | 63.2 |
| Flowers(Test) | **72.3** | 67.6 |
| Flowers | **75.5** | 58.6 |
| ImageNet-10 | 98.6 | **99.5** |
| Fashion | 64.2 | **70.0** |
| EuroSAT | 64.3 | **64.5** |
| ImageNet-Dogs | 38.9 | **84.4** |
| CIFAR-10 | 75.8 | **92.2** |
| OxfordPets | 52.4 | **72.7** |
| DTD | 46.5 | **47.4** |
| STL-10 | 94.4 | **98.3** |
| Food101 | 57.2 | **62.9** |
| CIFAR-100 | 43.0 | **59.0** |
| TinyImageNet | 41.1 | **46.6** |
| **AVG.** | 59.4 | **63.1** |

## K  COMPARISON OF DIFFERENT MODALITY CLUSTER HEADS

To evaluate the impact of different modality-specific cluster heads on the final clustering assignment. We compare three cluster heads: Visual Head (V-Head), Spatial Head (S-Head), and Textual Head (T-Head). For datasets where the textual feature was not selected ($\tau \leq 0.33$), the T-Head is unavailable (N/A). All cluster heads are trained via our mutual distillation framework.

Table 9 compares the clustering performance when using different modality cluster heads for final assignment: Visual Head (V-Head), Spatial Head (S-Head), and Textual Head (T-Head). The results robustly validate our design decision to use the distilled visual cluster head as the primary output. While the spatial and textual cluster heads can occasionally achieve comparable or even marginally better results on certain individual datasets (e.g., S-Head on GTSRB and TinyImageNet, T-Head on Flowers), the distilled visual cluster head demonstrates superior and more consistent performance on average across all benchmark datasets. This outcome confirms that the rich semantic representations from the CLIP visual encoder provide the most reliable foundation for final clustering. The spatial features serve as a valuable complementary source of structural information during the mutual distillation process, and the textual features, when beneficial ($\tau > 0.33$), act effectively as a "semantic teacher." However, the visual modality ultimately proves to be the most robust anchor for producing the final cluster assignments, achieving the highest average performance across all three evaluation metrics.

Table 9: Clustering performance using different modality cluster heads (ACC% / NMI% / ARI%). For datasets where textual feature were not selected ($\tau \leq 0.33$), the T-Head was not available (N/A). The best results for each dataset are in **Bold**.

| Dataset | V-Head | | | S-Head | | | T-Head | | |
|---------|--------|--------|--------|--------|--------|--------|--------|--------|--------|
| | ACC | NMI | ARI | ACC | NMI | ARI | ACC | NMI | ARI |
| MNIST | **89.8** | **87.0** | **83.6** | 89.5 | 86.7 | 83.3 | N/A | N/A | N/A |
| USPS | **85.8** | **80.6** | **73.4** | 77.9 | 81.3 | 72.0 | N/A | N/A | N/A |
| ImageNet-10 | **99.8** | **99.2** | **99.4** | 99.8 | 99.2 | 99.4 | N/A | N/A | N/A |
| GTSRB | 36.2 | 58.3 | 28.1 | **38.7** | **60.4** | **31.5** | N/A | N/A | N/A |
| Fashion | **76.0** | **71.2** | **62.8** | 72.4 | 70.4 | 61.3 | N/A | N/A | N/A |
| FER2013 | **31.5** | **9.5** | **7.3** | 27.7 | 7.6 | 5.3 | N/A | N/A | N/A |
| EuroSAT | **86.3** | **78.9** | **75.1** | 77.2 | 73.4 | 66.9 | N/A | N/A | N/A |
| ImageNet-Dogs | **91.4** | **89.7** | **86.7** | 89.4 | 87.1 | 83.0 | N/A | N/A | N/A |
| Resisc45 | **81.4** | **82.6** | **71.1** | 76.9 | 80.0 | 66.4 | N/A | N/A | N/A |
| OxfordPets | **89.4** | **90.9** | **82.6** | 82.9 | 86.2 | 73.4 | N/A | N/A | N/A |
| DTD | **57.4** | **63.4** | **40.4** | 50.3 | 61.0 | 35.9 | N/A | N/A | N/A |
| CIFAR-10 | **94.5** | **88.9** | **88.3** | 94.4 | 87.6 | 88.1 | 90.2 | 83.2 | 82.9 |
| Flowers(Test) | 67.1 | 85.9 | 63.2 | 68.1 | **86.1** | 61.4 | **68.6** | 85.4 | **63.0** |
| STL-10 | **99.0** | **97.3** | **97.9** | 98.9 | 97.1 | 97.6 | 97.6 | 94.3 | 94.8 |
| Food101 | **65.2** | **71.1** | **50.7** | 60.9 | 65.7 | 45.1 | 62.6 | 68.2 | 46.7 |
| CIFAR-100 | **63.4** | **70.5** | **48.1** | 56.4 | 68.8 | 43.3 | 54.1 | 64.9 | 39.1 |
| Flowers | 60.5 | 83.0 | 53.8 | 66.0 | **85.8** | 59.1 | **66.4** | 85.4 | **59.2** |
| TinyImageNet | 52.0 | 65.1 | 36.8 | **53.2** | **65.8** | **37.8** | 50.3 | 61.8 | 33.1 |
| AVG. | **73.7** | **76.3** | **63.8** | 71.1 | 75.0 | 61.7 | - | - | - |

## L   THEORETICAL GUIDANCE FOR UTILIZING TEXTUAL COMPACTNESS

**Theorem 1**  *Let $V$ (visual feature), $T$ (textual feature), and $Y \in \{1, 2, \ldots, K\}$ (classification label, $K \geq 2$) be random variables. Denote the Bayes classification error rate based on $(V, T)$ as $R_{V,T}^* = \min_g \mathbb{P}(g(V, T) \neq Y)$, where the minimum is taken over all measurable functions $g$. Define the function*

$$f(p) = H_{\mathrm{bin}}(p) + p \log(K - 1), \quad p \in [0, 1 - 1/K],$$

*where $H_{\mathrm{bin}}(p) = -p \log p - (1 - p) \log(1 - p)$ is the binary entropy (logarithms in base 2). Then,*

$$R_{V,T}^* \geq f^{-1}\big(H(Y|V) - H(T|V) + H(T|V, Y)\big),$$

*where $f^{-1}$ is the inverse of $f$ on $[0, 1 - 1/K]$, which is strictly increasing.*

**Proof 1**  *The $K$-class Fano inequality states that*

$$H(Y|V, T) \leq H_{\mathrm{bin}}(R_{V,T}^*) + R_{V,T}^* \log(K - 1) = f(R_{V,T}^*).$$

*From the definition of conditional mutual information,*

$$I(T; Y|V) = H(Y|V) - H(Y|V, T),$$

*and also*

$$I(T; Y|V) = H(T|V) - H(T|V, Y).$$

*Equating the two expressions yields*

$$H(Y|V) - H(Y|V, T) = H(T|V) - H(T|V, Y),$$

*which rearranges to*

$$H(Y|V, T) = H(Y|V) - H(T|V) + H(T|V, Y).$$

*Substituting into the Fano inequality gives*

$$H(Y|V) - H(T|V) + H(T|V, Y) \leq f(R_{V,T}^*).$$

*The derivative of $f$ is*

$$f'(p) = \log \frac{1-p}{p} + \log(K-1) = \log\left((K-1) \cdot \frac{1-p}{p}\right).$$

*For $p \in (0, 1 - 1/K)$, we have $(1-p)/p > 1/(K-1)$, so $f'(p) > 0$. At the endpoints, $f$ is continuous and strictly increasing on $[0, 1 - 1/K]$. Since $R^*_{V,T} \in [0, 1 - 1/K]$, the inverse $f^{-1}$ exists and is strictly increasing on the range of $f$. Applying $f^{-1}$ to both sides of the inequality completes the proof.*

This theorem provides an information-theoretic lower bound on the Bayes error rate in multimodal classification tasks. In particular, when the textual feature $T$ exhibits higher uncertainty (high $\tau$ value) given the visual feature $V$ (i.e., a larger $H(T|V)$), the theoretical lower bound of the Bayes error rate $R^*_{V,T}$ becomes smaller, indicating a higher potential upper bound for classification performance.

## M  Evaluating the Contribution of ResNet-50 to Model Performance

CLIP could extract both image-level and patch-level features. However, since it is primarily optimized for image-text alignment in a shared semantic space, its representations tend to emphasize global semantics at the potential cost of localized spatial details. To address this, we introduce spatial features in which ResNet-50 extracts local spatial information from image patches, and GAT further captures relational dependencies among these patches.

In order to further analyze whether the performance improvement stems from the method design or from leveraging more powerful pre-trained features, we conducted a comprehensive ablation study comparing the following configurations: (1) ResNet-50 only: K-Means on features from the ImageNet-pretrained ResNet-50. (2) CLIP only: K-Means on CLIP visual features (our baseline). (3) Simple Fusion (CLIP + ResNet-50): K-Means on concatenated features from CLIP and the ImageNet-pretrained ResNet-50. (4) CLIP + Spatial Features: K-Means on concatenated CLIP visual features and our proposed spatial features. (5) CLIP + ResNet-50 + Distillation: Our framework uses CLIP visual features and ResNet-50 features with mutual distillation. (6) Full SATC (Ours): The complete pipeline.

Table 10 reports the results of this ablation study. The results presented in Table 10 clearly demonstrate that: (1) Using ResNet-50 features alone (Column 1) performs significantly worse than using CLIP features alone (Column 2). The average accuracy drops from 60.8% to 51.3%. (2) Simply concatenating CLIP and ResNet-50 features (Column 3) provides negligible improvement over CLIP alone (59.4% vs. 60.8%), and even degrades performance on several datasets. This indicates that naively adding more pre-trained features is not beneficial. (3) In contrast, using our spatial features with CLIP (Column 4) already yields a substantial gain (64.0%), confirming the value of the structural information captured by our encoder. (4) The full SATC pipeline (Column 6) achieves the best performance (73.7%). The step-by-step improvement from Column 2 to Column 4 to Column 6 demonstrates that the performance gain is primarily attributable to our novel integration of spatial structure and selective textual guidance, rather than the mere use of additional pre-trained features. These results provide evidence that the improvement is indeed a consequence of our method's design.

Table 10: ACC Results of Ablation study on the Effects of integrating ResNet-50

|  | 1 | 2 | 3 | 4 | 5 | 6 |
|---|---|---|---|---|---|---|
| CLIP |  | √ | √ | √ | √ | √ |
| ResNet-50 | √ |  | √ | √ | √ | √ |
| Distillation |  |  |  |  | √ | √ |
| GAT |  |  |  | √ |  | √ |
| 1 | 50.1 | 56.5 | 56.8 | 57.5 | 75.2 | **89.8** |
| 2 | 61.4 | 62.3 | 61.5 | 62.7 | 78.2 | **85.8** |
| 3 | 97.0 | 98.6 | 98.8 | 99.4 | 99.6 | **99.8** |
| 4 | 26.7 | 31.6 | 32.2 | 33.6 | 34.7 | **36.2** |
| 5 | 58.8 | 64.2 | 64.3 | 64.4 | 72.5 | **76.0** |
| 6 | 26.4 | 27.0 | 27.2 | 27.8 | 30.7 | **31.5** |
| 7 | 56.8 | 64.3 | 64.6 | 64.7 | 81.6 | **86.3** |
| 8 | 33.3 | 38.9 | 40.0 | 74.5 | 84.1 | **91.4** |
| 9 | 54.3 | 67.4 | 67.7 | 64.3 | 78.3 | **81.4** |
| 10 | 47.8 | 52.4 | 55.0 | 72.7 | 75.1 | **89.4** |
| 11 | 39.7 | 46.5 | 44.5 | 48.3 | 48.5 | **57.4** |
| 12 | 55.3 | 92.2 | 76.2 | 86.3 | 94.0 | **94.5** |
| 13 | 66.9 | 67.6 | 70.9 | **72.6** | 67.2 | 69.7 |
| 14 | 88.9 | 98.3 | 94.5 | 98.1 | 98.7 | **99.0** |
| 15 | 42.8 | 62.9 | 56.5 | 56.8 | 61.4 | **65.2** |
| 16 | 25.0 | 59.0 | 42.2 | 47.0 | 59.3 | **63.4** |
| 17 | 67.5 | 58.6 | 74.7 | **74.9** | 55.3 | 60.5 |
| 18 | 25.2 | 46.6 | 41.1 | 45.5 | 49.3 | **52.0** |
| Avg. | 51.3 | 60.8 | 59.4 | 64.0 | 69.1 | **73.7** |

