# OpenReview forum: "Spatial Structure and Selective Text Jointly Facilitate Image Clustering"
_ICLR.cc/2026/Conference — ICLR 2026 Poster_

### Official Review · Reviewer_nD9y · 2025-10-17

**Soundness:** 2
**Presentation:** 2
**Contribution:** 2
**Rating:** 4
**Confidence:** 4

**Summary:**

This paper studies the task of image clustering. It introduces a framework named **Spatial Structure and Selective Text Jointly Facilitate Image Clustering, a novel framework desigend to overcome the limitations of existing CLIP-based deep clustering methods, particularly regarding their representation of spatial structures and indiscriminate use of text features.**

The framework fuses visual, spatial, and selectively chosen text features using a designed Tri-modal mutual distillation strategy.

Extensive experiments across 18 benchmarks show SATC’s good performance and effectiveness.

**Strengths:**

1. Superior Clustering Performance: The proposed method consistently achieved highest clustering results compared to extensive prior works across 18 benchmarks, showing substantial improvements.
2. The idea of incorporating spatial structure is interesting and effective.
3. The idea of textual selection is also insightful
4. Efficiency and Scalability: SATC not only achieves higher clustering accuracy but also maintains competitive or even lower running times compared to the TAC baseline across most datasets

**Weaknesses:**

**Potential Flag:** The Use of Large Language Models (LLMs) is **not disclosed** in the current manuscript, which is a violation of the new rule imposed by ICLR this year.

**W1:** Experiments on novel, unseen datasets are needed. All evaluated datasets in the current work might be explicitly leveraged during CLIP training. Therefore, it is hard to confirm the effectiveness of the proposed framework without control experiments on complete unseen, novel images. It is practical and important because clustering methods are often used to explore and understand unseen, novel images without labels.

**W2:** The authors discussed the pros and cons of visual and textual modalities in CLIP, and their effects on image clustering. Why not compare with DINOv2, v3 for image clustering under the same model size? A comparison with DINOv2 and v3 that uses a single modality is necessary. KMeans + DINOv2 / v3 features is a good baseline.

**W3:** The textual feature selector and tri-modal objective function are both depent on empirically set threshold $\tau$ and $\alpha$. How do the authors select these hyperparameters? What are the selection criteria or dataset? The authors mentioned it is “based on extensive experiments” at Line#331,  however, If all parameters tuned on the  test set, it is unfair.

If the above primary concerns could be addressed during the discussion stage, the reviewer is open to rise the rating.

**Questions:**

**Q1:** The Tri-modal mutual distillation framework utilizes three loss types: distillation ($L_{distill}$), consistency ($L_{consist}$ with $\lambda_1=1.0$), and entropy ($L_{entropy}$ with $\lambda_2=5.0$). What specific theoretical or empirical justification underpins the choice to set the **weight of the entropy loss ($\lambda_2$) five times higher** than the consistency loss ($\lambda_1$)?

**Q2:** The final cluster assignments are consistently produced by the **distilled visual cluster head**. How would the clustering performance be affected if the final assignment were instead derived from the distilled spatial cluster head or the distilled textual cluster head, especially given that mutual distillation is shown to be superior overall?

---

> ### Author Response · Authors · 2025-11-20
>
> ### Response to Reviewer nD9y
>
> ---
>
> We are very grateful for your valuable comments and questions. We try our best to address your questions as follows.
>
> ---
>
> **To Potential Flag (About Use of LLMs):** We used LLM solely for grammar checking. Since LLM provided no substantive intellectual contribution, in accordance with ICLR guidelines, no LLM contribution statement is included.
>
> **To W1 (About Generalization to CLIP-Unseen Data):** Good comments. To address the concern about potential data contamination, we would like to clarify that while CLIP was trained on a large-scale dataset, its pretraining corpus does not comprehensively cover all benchmarks used in our evaluation. In fact, several datasets included in our study—such as MNIST, EuroSAT, GTSRB, FER2013, and Flowers—are explicitly listed in the CLIP paper as not being part of its training data [1]. These can therefore be reasonably regarded as novel and unseen, supporting a valid assessment of our framework's generalization capability.
>
> Furthermore, even for datasets that may have been explicitly included in CLIP’s pretraining, directly applying K-Means on CLIP-extracted features does not yield strong clustering performance. This highlights that our proposed framework is necessary to improve clustering quality. The corresponding results are reported in Appendix B, Table 5.
>
> [1] Learning Transferable Visual Models From Natural Language Supervision, ICML 2021.
>
> ---
>
> **To W2 (About Comparison with DINOv2):** Nice suggestion. Our work primarily focuses on enhancing CLIP's representation capability, then DINOv2 is not compared. KMeans + DINOv2 features is indeed a good baseline.
>
> We have included KMeans with DINOv2 (ViT-B/14) features as an additional baseline. As shown in the following table, our method remains highly competitive and outperforms DINOv2 across most benchmarks. *The results of the newly added baselines have been incorporated into Table 1 of the manuscript.*
>
> |Dataset|DINOv2+KMeans|SATC|
> |-|-|-|
> |MNIST|54.1|**89.8**|
> |USPS|60.8|**85.8**|
> |ImageNet-10|96.3|**99.8**|
> |GTSRB|22.7|**36.2**|
> |Fashion|69.6|**76.0**|
> |FER2013|30.0|**31.5**|
> |EuroSAT|62.3|**86.3**|
> |ImageNet-Dogs|87.5|**91.4**|
> |Resisc45|62.6|**81.4**|
> |OxfordPets|82.2|**89.4**|
> |DTD|53.6|**57.4**|
> |CIFAR-10|77.0|**94.5**|
> |Flowers(Test)|**85.2**|67.1|
> |STL-10|77.4|**99.0**|
> |Food101|64.9|**65.2**|
> |CIFAR-100|62.5|**63.4**|
> |Flowers|**90.3**|60.5|
> |TinyImageNet|**67.5**|52.0|
> |**Avg.**|67.0|**73.7**|
>
> ---
>
> **To W3 and Q1 (About Parameter Setting):** We thank the reviewer for raising this critical methodological point.
>
> The loss weights $\lambda_1=1.0$ and $\lambda_2=5.0$ were initially adopted from TAC [2]. We further conducted systematic parameter analysis as presented in Section 4.4 and Figure 4, which confirms that this ratio delivers optimal performance across multiple benchmark datasets. The higher value of $\lambda_2$ effectively prevents cluster collapse by encouraging more balanced assignment distributions.
>
> Regarding the compactness threshold (τ), we would like to clarify that τ = 0.33 is best understood as an empirically discovered rule derived from broad benchmarking, rather than a hyperparameter tuned on test performance. This value was identified based on aggregated trends observed across all 18 datasets in our study. Importantly, it was not iteratively optimized for any specific dataset's test performance. The consistent effectiveness of this threshold across such diverse benchmarks suggests it captures a generalizable pattern in textual feature utility, rather than representing a statistic overfitted to particular test sets.
>
> [2] Image Clustering with External Guidance, ICML, 2024.
>
> ---
>
> **To Q2 (About the Choice of Visual Head for Final Assignment):** Thanks for the insightful question. Since both spatial and textual features serve as auxiliary information in our image clustering framework, we use the cluster assignments from the distilled visual head as the final output. As suggested, we have evaluated all three distilled cluster heads in terms of ACC, with results shown in the table below. The visual head demonstrates superior and more consistent performance on average across all benchmark datasets, confirming its suitability as the primary output modality. *The relevant experimental results have been added to Appendix K of the manuscript.*
>
> |Dataset|V-Head|S-Head|T-Head|
> |-|-|-|-|
> |MNIST|**89.8**|89.5|N/A|
> |USPS|**85.8**|77.9|N/A|
> |ImageNet-10|**99.8**|99.8|N/A|
> |GTSRB|36.2|**38.7**|N/A|
> |Fashion|**76.0**|72.4|N/A|
> |FER2013|**31.5**|27.7|N/A|
> |EuroSAT|**86.3**|77.2|N/A|
> |ImageNet-Dogs|**91.4**|89.4|N/A|
> |Resisc45|**81.4**|76.9|N/A|
> |OxfordPets|**89.4**|82.9|N/A|
> |DTD|**57.4**|50.3|N/A|
> |CIFAR-10|**94.5**|94.4|90.2|
> |Flowers(Test)|67.1|68.1|**68.6**|
> |STL-10|**99.0**|98.9|97.6|
> |Food101|**65.2**|60.9|62.6|
> |CIFAR-100|**63.4**|56.4|54.1|
> |Flowers|60.5|66.0|**66.4**|
> |TinyImageNet|52.0|**53.2**|50.3|
> |**AVG.**|**73.7**|71.1|-|
>
> ---

---

> > ### Comment · Reviewer_nD9y · 2025-11-25
> >
> > Thanks for providing the rebuttal response.
> >
> > From your response, my concerns are partially addressed. However, some of my concerns still exist.
> >
> > - As you mention in your response, "Regarding the compactness threshold (τ), we would like to clarify that τ = 0.33 is best understood as an empirically discovered rule derived from broad benchmarking, rather than a hyperparameter tuned on test performance. This value was identified based on aggregated trends observed across all 18 datasets in our study.", the prpposed method relies on emperical parameter setting. For this, I would like to ask, where is this "best understood" from? based on what experience that from what datasets? How did the authors confirm this as "best"?
> >
> > - Beside, how is the performance compared to simple DINOv3 vs. KMeans?
> >
> > Thus, I remain my rating of 4 for the manuscript mainly because the framework seems rely heavily on hyperparameter settings.

---

> > > ### Author Response · Authors · 2025-11-27
> > >
> > > We are very grateful for your reply.
> > >
> > > * We apologize that the expression "is best understood" might cause some confusion. One of the fundamental contributions of our work lies not in the specific threshold value, but in the conceptual proposition that the utility of textual features is dataset-dependent, together with a quantitative metric—compactness (τ)—to assess it. We evaluated τ values across all 18 datasets and found that τ = 0.33 effectively captures the transition point beyond which textual features shift from being redundant or misleading to consistently beneficial for clustering performance. The value τ = 0.33 was identified based on aggregated trends observed across the 18 datasets in our study (Table 2). It was not optimized for specific dataset's test performance.
> > >
> > > * Following your recommendation, we attempted to incorporate DINOv3 into our experiments. However, The DINOv3 paper was released shortly before the submission of our work, and the authors of DINOv3 have not publicly released the pretrained model, making it infeasible for us to directly include it as a baseline at this time. Then, we have added DINOv2 + K-Means as an additional baseline during the rebuttal phase.
> > >
> > > * Furthermore, our method can be readily extended to a method that avoids the compactness threshold by adopting a weighting strategy. To demonstrate this flexibility, we have designed and conducted additional experiments using a linear weighting scheme for the textual modality based on τ. The results, summarized in the table below, confirm the effectiveness of the method.
> > >
> > > | Dataset | TAC  | Weighted |
> > > |-|-|-|
> > > | MNIST | 39.8 | **90.1** |
> > > | USPS | 55.2 | **84.3** |
> > > | ImageNet-10 | 99.5 | **99.8** |
> > > | GTSRB | 27.7 | **38.5** |
> > > | Fashion | 70.0 | **75.4** |
> > > | FER2013 | 25.4 | **32.2** |
> > > | EuroSAT  | 64.5 | **83.2** |
> > > | ImageNet-Dogs | 84.4 | **91.7** |
> > > | Resisc45 | 63.2 | **75.1** |
> > > | OxfordPets | 72.7 | **86.5** |
> > > | DTD | 47.4 | **51.8** |
> > > | CIFAR-10 | 92.2 | **94.4** |
> > > | Flowers(Test) | 67.6 | **72.0** |
> > > | STL-10 | 98.3 | **98.9** |
> > > | Food101 | 62.9 | **63.7** |
> > > | CIFAR-100 | 59.0 | **60.2** |
> > > | Flowers | 58.6 | **53.9** |
> > > | TinyImageNet | 46.6 | **52.7** |
> > > | **AVG.** | 63.1 | **72.5** |

---

> ### Comment · Reviewer_nD9y · 2025-11-27
>
> Thank you for the further clairifcation.
>
> - Please note that DINOv3 weights have been publicly available here 3 months ago: https://github.com/facebookresearch/dinov3
>
> - From your answer "The value τ = 0.33 was identified based on aggregated trends observed across the 18 datasets", it is clear and obvious the value of τ is optimized globally for the 18 datasets. Even though it is not optimized for each dataset, it is optimized together for these datasets. If you conduct such optimization, you need to conduct this optimization for parameters or settings for every method that you compare. Otherwise, it is not a fair comparison.
>
> Thus, I further confirm my rating of 4.

---

> ### Author Response · Authors · 2025-12-02
>
> Thank you for your further communication.
>
> ---
>
> **About Comparison with DINOv3**
>
> Thank you very much for the reviewer’s helpful pointer to the public release of DINOv3 weights. Following the reviewer’s suggestion, we have successfully incorporated DINOv3 into our benchmark.
>
> Specifically, we selected the DINOv3-ViT-B/16 model (343M), whose architectural scale is comparable to CLIP-ViT-B/32 (330M), and extracted image features for 18 datasets using the officially released pretrained weights. We then applied KMeans under the same clustering settings as in our other baselines. As shown in the table below, the performance of DINOv3 is consistent with the strong representation capability reported in the original paper, and our method continues to outperform this new baseline across datasets.
>
> |Dataset|DINOv2(337M)|DINOv3(343M)|SATC(330M)|
> |-|-|-|-|
> |MNIST|54.1|60.3|**89.8**|
> |USPS|60.8|63.4|**85.8**|
> |ImageNet-10|96.3|96.7|**99.8**|
> |GTSRB|22.7|30.2|**36.2**|
> |Fashion|69.6|65.9|**76.0**|
> |FER2013|30.0|30.6|**31.5**|
> |EuroSAT|62.3|67.1|**86.3**|
> |ImageNet-Dogs|87.5|48.3|**91.4**|
> |Resisc45|62.6|55.0|**81.4**|
> |OxfordPets|82.2|55.6|**89.4**|
> |DTD|53.6|55.9|**57.4**|
> |CIFAR-10|77.0|90.1|**94.5**|
> |Flowers(Test)|85.2|**85.9**|67.1|
> |STL-10|77.4|98.1|**99.0**|
> |Food101|64.9|48.8|**65.2**|
> |CIFAR-100|62.5|57.6|**63.4**|
> |Flowers|90.3|**91.1**|60.5|
> |TinyImageNet|**67.5**|52.3|52.0|
> |**Avg.**|67.0|64.1|**73.7**|
>
> ---
>
> **About Comparison Fairness**
>
> We agree that a fair comparison is important. **Ensuring fairness in comparison is a fundamental principle that we adhere to.**
>
> For the competing methods (except those involving K-means), we adopted their reported results, and the parameters follow the recommended settings provided by the authors.
>
> Most of these methods have undergone parameter sensitivity analysis on the test data and **employ optimized settings**, such as the temperature hyperparameters κ and τ in GradNorm and the lower confidence bound α for pseudo labels in TCL. In addition, some methods set different parameters for different datasets **to achieve optimal performance**, such as the noise intensity σ in LFSS. Table 1 presents the optimally tuned performance for each method. In contrast, our method keeps the textual compactness threshold strictly identical across all datasets; **this supports the fairness of the evaluation** and validates our hypothesis regarding the dataset-dependent usefulness of textual features.
>
> Moreover, we would like to further clarify that the textual compactness τ is not a tunable parameter for performance improvement, but rather a measure of the intra-cluster compactness of textual features for the dataset. The textual compactness threshold (0.33) reflects a universally observed transition point where textual signals shift from hindering to enhancing image clustering (Table 2).
>
> ---

---

### Official Review · Reviewer_eiRh · 2025-10-21

**Soundness:** 3
**Presentation:** 2
**Contribution:** 2
**Rating:** 4
**Confidence:** 5

**Summary:**

Based on the externally guided clustering paradigm, this paper further leverages spatial information to distinguish visually and textually similar instances. The proposed method is extensively evaluated on 18 image clustering datasets, demonstrating superior performance over previous studies.

**Strengths:**

1. The contribution of this work is clear, i.e., leveraging the spatial feature in addition to visual and textual features to facilitate image clustering. Such a motivation is straightforward.
2. Extensive experiments across 18 datasets demonstrate the effectiveness of the proposed method.
3. The ablation study on incorporating textual semantics with the compactness metric is interesting.

**Weaknesses:**

1. The writing in section 3.1 is confusing. Where exactly is the graph attention applied? On different images, or on patches within a single image?
2. Besides the pre-trained CLIP model, a pre-trained ResNet-50 model is also utilized in the proposed method. It is questionable why ResNet-50 is needed, since CLIP could already extract both image- and patch-level features. Does the performance improvement of the proposed method come from introducing the ResNet model?
3. How are the textual compactness metrics in Eq. 5 used? It should be explained more clearly in the subsection.
4. Since the proposed SATC is more efficient than TAC, experimental results on the full ImageNet-1K are expected.

**Questions:**

My major concerns lie in whether the performance gain comes from the proposed method additionally leverages a pre-trained ResNet-50 model. Besides, some details of the proposed method should be explained more clearly. I will raise my score if my concerns are well addressed.

---

> ### Author Response · Authors · 2025-11-20
>
> ### Response to Reviewer eiRh
>
> ---
>
> We are very grateful for your valuable comments and questions. We try our best to address your questions as follows.
>
> ---
>
> **To W1 (Clarification on Section 3.1):** Sorry for this confusion. The GAT is applied to patches within a single image to capture spatial relationships between different regions of the same image. For each individual image, we extract patch-level features and construct a graph where nodes represent image patches and edges connect semantically related patches within that image. The GAT then operates on this intra-image graph to model relational dependencies between different patches, capturing structural information about object parts and their spatial arrangements. This process is repeated independently for each image in the dataset. We have revised Section 3.1 to clarify this point.
>
> ---
>
> **To W2 and Q (About the Use of Pre-trained ResNet-50):**
>
> Nice comment. Yes, CLIP could extract both image- and patch-level features. However, as it is primarily optimized for image-text alignment in a shared semantic space, its representations tend to emphasize global semantics at the potential cost of localized spatial details. This limitation is visually supported in Section 4.3.3 and Figure 3.
>
> To address this, we introduce spatial features, in which ResNet-50 extracts local spatial information from image patches, and GAT further captures relational dependencies among these patches. We have conducted ablation studies to evaluate the contribution of each component, with results summarized in the table below:
>
> (1) Contribution of ResNet-50 features. Comparing the first two columns, incorporating ResNet-50 node features consistently improves clustering accuracy across datasets. This confirms that ResNet-50 captures valuable patch-level cues that are underrepresented in CLIP features, aligning with our expectations. (2) Contribution of spatial modeling with GAT. Introducing spatial modeling on ResNet-50 node features using GAT further boosts clustering performance. This improvement arises from GAT’s adaptive attention mechanism.
>
> In conclusion, the performance improvement stems from the combined effect of ResNet-50's local feature extraction and GAT's relational modeling. *The relevant experimental results have been added to Section 4.3.2 of the manuscript.*
>
> |CLIP|√|√|√|
> |-|-|-|-|
> |ResNet-50| |√|√|
> |GAT| | |√|
> |1|56.5|75.2|**89.8**|
> |2|62.3|78.2|**85.8**|
> |3|98.6|99.6|**99.8**|
> |4|31.6|34.7|**36.2**|
> |5|64.2|72.5|**76.0**|
> |6|27.0|30.7|**31.5**|
> |7|64.3|81.6|**86.3**|
> |8|38.9|84.1|**91.4**|
> |9|67.4|78.3|**81.4**|
> |10|52.4|75.1|**89.4**|
> |11|46.5|48.5|**57.4**|
> |12|92.2|94.0|**94.5**|
> |13|67.6|67.2|**69.7**|
> |14|98.3|98.7|**99.0**|
> |15|62.9|61.4|**65.2**|
> |16|59.0|59.3|**63.4**|
> |17|58.6|55.3|**60.5**|
> |18|46.6|49.3|**52.0**|
> |**Avg.**|60.8|69.1|**73.7**|
>
> ---
>
> **To W3 (About the Utilization of Eq.5):** We apologize for the lack of clarity. The textual compactness τ, defined in Eq. 5, is used as a dataset-level prior. Before training the tri-modal distillation framework, we compute τ on the entire dataset's textual features. If τ is higher than a threshold, we consider the textual modality to be beneficial and include it in the mutual distillation process; otherwise, we exclude it. This is a one-time, pre-training decision that efficiently prevents the model from being misled by unhelpful textual signals. *We have revised Section 3.2.2 to make this pipeline clearer.*
>
> ---
>
> **To W4 (About Experiments on ImageNet-1K):**
>
> We have conducted clustering experiments on the full ImageNet-1K dataset. The results are shown in the following table, where SATC outperforms TAC across all metrics (ACC, NMI, ARI).
>
> |ImageNet-1K|TAC|SATC|
> |-|-|-|
> |ACC%|58.9|**63.9**|
> |NMI%|80.1|**82.7**|
> |ARI%|43.9|**46.3**|
>
> ---

---

> > ### Comment · Reviewer_eiRh · 2025-11-26
> >
> > Thanks for the responses. Most of my concerns have been addressed. However, since the proposed method additionally uses the pre-trained ResNet50 (trained on ImageNet?), the performance comparisons with existing baselines could be unfair. More evidences are expected to demonstrate that the superior performance comes from the method design instead of more discriminative pre-trained features (about 10% improvement).

---

> ### Author Response · Authors · 2025-11-27
>
> We are very grateful for your reply.
>
> We sincerely thank the reviewer for raising this important point regarding the fairness of comparisons. We agree that it is essential to ensure performance gains stem from our methodological design rather than simply leveraging more powerful pre-trained features.
>
> To rigorously isolate the source of improvement, we have conducted a comprehensive ablation study comparing the following configurations:
>
> 1. ResNet-50 only: K-Means on features from the ImageNet-pretrained ResNet-50.
>
> 2. CLIP only: K-Means on CLIP visual features (our baseline).
>
> 3. Simple Fusion (CLIP + ResNet-50): K-Means on concatenated features from CLIP and the ImageNet-pretrained ResNet-50.
>
> 4. CLIP + Spatial Features: K-Means on concatenated CLIP visual features and our proposed spatial features.
>
> 5. CLIP + ResNet-50 + Distillation: Our framework using CLIP visual features and ResNet-50 features with mutual distillation.
>
> 6. Full SATC (Ours): The complete pipeline.
>
> The results, presented in the table below, clearly demonstrate that:
>
> 1. Using ResNet-50 features alone (Column 1) performs significantly worse than using CLIP features alone (Column 2). The average accuracy drops from 60.8% to 51.3%.
>
> 2. Simply concatenating CLIP and ResNet-50 features (Column 3) provides negligible improvement over CLIP alone (59.4% vs. 60.8%), and even degrades performance on several datasets. This indicates that naively adding more pre-trained features is not beneficial.
>
> 3. In contrast, using our spatial features with CLIP (Column 4) already yields a substantial gain (64.0%), confirming the value of the structural information captured by our encoder.
>
> 4. The full SATC pipeline (Column 6) achieves the best performance (73.7%). The step-by-step improvement from Column 2 to Column 4 to Column 6 demonstrates that the performance gain is primarily attributable to our novel integration of spatial structure and selective textual guidance, rather than the mere use of additional pre-trained features.
>
> These results provide strong evidence that the improvement is indeed a consequence of our method's design.
>
> | |1|2|3|4|5|6|
> |-|-|-|-|-|-|-|
> |CLIP| |√|√|√|√|√|
> |ResNet-50| √| |√|√|√|√|
> |Distillation| | | | |√|√|
> |GAT| | | |√| |√|
> |1|50.1|56.5|56.8|57.5|75.2|**89.8**|
> |2|61.4|62.3|61.5|62.7|78.2|**85.8**|
> |3|97.0|98.6|98.8|99.4|99.6|**99.8**|
> |4|26.7|31.6|32.2|33.6|34.7|**36.2**|
> |5|58.8|64.2|64.3|64.4|72.5|**76.0**|
> |6|26.4|27.0|27.2|27.8|30.7|**31.5**|
> |7|56.8|64.3|64.6|64.7|81.6|**86.3**|
> |8|33.3|38.9|40.0|74.5|84.1|**91.4**|
> |9|54.3|67.4|67.7|64.3|78.3|**81.4**|
> |10|47.8|52.4|55.0|72.7|75.1|**89.4**|
> |11|39.7|46.5|44.5|48.3|48.5|**57.4**|
> |12|55.3|92.2|76.2|86.3|94.0|**94.5**|
> |13|66.9|67.6|70.9|**72.6**|67.2|67.1|
> |14|88.9|98.3|94.5|98.1|98.7|**99.0**|
> |15|42.8|62.9|56.5|56.8|61.4|**65.2**|
> |16|25.0|59.0|42.2|47.0|59.3|**63.4**|
> |17|67.5|58.6|74.7|**74.9**|55.3|60.5|
> |18|25.2|46.6|41.1|45.5|49.3|**52.0**|
> |Avg.|51.3|60.8|59.4|64.0|69.1|**73.7**|

---

> ### Comment · Reviewer_eiRh · 2025-11-28
>
> Thanks for the further clarification. Now my concerns have been addressed and I would like to raise my score accordingly.

---

> > ### Author Response · Authors · 2025-12-03
> >
> > Thank you for your confirmation and for raising the score. We appreciate your constructive review.

---

### Official Review · Reviewer_GM72 · 2025-11-01

**Soundness:** 4
**Presentation:** 3
**Contribution:** 2
**Rating:** 6
**Confidence:** 4

**Summary:**

The work combines three different modalities (visual, spatial, textual) to enhance clustering performance on a variety of image datasets. One key contribution is the newly introduced spatial modality that encodes relationships between image patches. To effectively leverage the different modalities a new framework is introduced to enforce cross-modal alignment in image clustering. Additionally, the authors establish an adaptive textual feature selector that estimates the benefits of using textual features during clustering. This prevents performance degradation on datasets where textual descriptions are uninformative or misleading. The experiments report SOTA performance across the vast majority of the 18 datasets.

**Strengths:**

1. The novel textual feature extractor is well motivated and proves to be of great benefit to the clustering performance
2. Comprehensive empirical validations were made on various datasets
3. SOTA results on a vast majority of datasets

**Weaknesses:**

1. The results are missing standard deviations to estimate the actual statistical significance of the proposed method
2. Are spatial features really a contribution or could visual-textual be sufficient? An ablation studies on the impact of the addition of spatial features for clustering would be great.
3. The compactness metric threshold appears to be found through exhaustive search rather than principled derivation
4. Typo in 324/328.

**Questions:**

1. Please provide the standard deviations for all datasets to estimate the statistical significance of the reported improvements.
2. What specific relational dependencies do the spatial features capture that CLIP's ViT doesn't already encode?
3. How was the 0.33 threshold determined? Based on the train/val split or post-hoc on the test data? Why does the usage of textual features need to be a binary decision rather than being modeled by weights determining their impact?

---

> ### Author Response · Authors · 2025-11-17
>
> ### Response to Reviewer GM72
>
> ---
>
> We are very grateful for your valuable comments and questions. We try our best to address your questions as follows.
>
> ---
>
> **To W1 and Q1 (About Standard Deviations):**
>
> Thank you for this suggestion. We have now incorporated the standard deviations for all experimental results. All performance metrics are reported as mean ± standard deviation over ten independent runs. Statistically significant improvements of SATC over the TAC baseline are marked with an asterisk (*), determined by a paired t-test with p < 0.05.
>
> |Dataset|TACACC|SATCACC|TACNMI|SATCNMI|TACARI|SATCARI|
> |-|-|-|-|-|-|-|
> |MNIST|39.8±1.3|**89.8±0.2***|31.5±1.2|**87.0±0.2***|22.3±1.2|**83.6±0.1***|
> |USPS|55.2±3.0|**85.8±2.3***|51.0±1.0|**80.6±1.3***|44.0±2.4|**73.4±2.5***|
> |ImageNet-10|99.5±0.0|**99.8±0.0***|98.5±0.0|**99.2±0.0***|98.8±0.0|**99.4±0.0***|
> |GTSRB|27.7±0.8|**36.2±1.0***|43.7±1.1|**58.3±0.6***|18.9±0.8|**28.1±0.9***|
> |Fashion|70.0±2.5|**76.0±3.2***|66.2±2.7|**71.2±2.5***|56.7±3.4|**62.8±3.3***|
> |FER2013|25.4±1.0|**31.5±1.2***|6.0±1.0|**9.5±0.8***|4.4±0.7|**7.3±0.8***|
> |EuroSAT|64.5±3.8|**86.3±6.5***|53.3±2.8|**78.9±4.7***|50.0±3.4|**75.1±7.3***|
> |ImageNet-Dogs|84.4±1.9|**91.4±0.7***|77.4±1.1|**89.7±0.8***|72.0±1.5|**86.7±1.1***|
> |Resisc45|63.2±0.8|**81.4±1.3***|70.9±0.8|**82.6±0.6***|50.0±1.0|**71.1±1.0***|
> |OxfordPets|72.7±1.1|**89.4±1.3***|79.1±0.9|**90.9±0.7***|61.2±1.3|**82.6±1.4***|
> |DTD|47.4±0.9|**57.4±1.1***|57.3±0.6|**63.4±0.6***|32.1±0.9|**40.4±0.9***|
> |CIFAR-10|92.2±1.6|**94.5±0.1***|83.7±1.1|**88.9±0.1***|83.6±1.8|**88.3±0.1***|
> |Flowers(Test)|**67.6±0.9**|67.1±1.4|84.3±0.3|**85.9±0.4***|**63.4±1.4**|63.2±1.2|
> |STL-10|98.3±0.0|**99.0±0.0***|95.7±0.1|**97.3±0.1***|96.3±0.1|**97.9±0.1***|
> |Food101|62.9±0.9|**65.2±1.3***|69.0±0.6|**71.1±0.4***|47.4±0.8|**50.7±0.9***|
> |CIFAR-100|59.0±0.9|**63.4±1.0***|68.6±0.4|**70.5±0.3***|42.7±0.6|**48.1±0.8***|
> |Flowers|58.6±2.0|**60.5±1.8***|**83.2±0.5**|83.0±0.5|**54.1±1.5**|53.8±1.1|
> |TinyImageNet|46.6±0.4|**52.0±0.6***|61.4±0.3|**65.1±0.2***|31.6±0.4|**36.8±0.5***|
>
> ---
>
> **ToW2 (Regarding Ablation Study on Spatial Features):**
>
> The contribution of spatial features is evaluated through two complementary experimental setups in our work:
> (1) Across all datasets, we compare the performance of using only visual features (+V) against combining visual and spatial features (+V+S). As reported in Section 4.3.3 and illustrated in Figure 3, the consistent improvement observed with (+V+S) confirms that spatial features provide meaningful and effective complementary information.
> (2) Furthermore, on the 7 datasets where textual features are beneficial (i.e., where τ > 0.33), we compare the visual-textual baseline (+V+T) against our complete SATC framework (+V+T+S). Results presented in the lower section of Table 4 demonstrate that incorporating spatial features yields additional performance gains beyond what is achieved by visual and textual modalities alone, underscoring the additive value of spatial information even in a multi-modal setting.
>
> ---

---

> ### Author Response · Authors · 2025-11-20
>
> ---
>
> **To W3 and Q3 (About the Compactness Threshold):** We thank the reviewer for this astute observation. You are right that the compactness threshold was determined empirically. However, the fundamental contribution of our work lies not in the specific threshold value, but in the conceptual proposition that the utility of textual features is dataset-dependent, coupled with a quantitative metric—compactness (τ)—to assess it. We would like to clarify that τ = 0.33 is understood as an empirically observed rule derived from broad benchmarking, rather than a hyperparameter tuned on test performance. This value was identified based on aggregated trends observed across all 18 datasets in our study (Table 2). Importantly, it was not iteratively optimized for any specific dataset's test performance.
>
> The suggestion of using a continuous weight is excellent. In this work, to directly validate our core hypothesis regarding the dataset-dependent applicability of textual features, we specifically adopted a binary decision mechanism.
>
> We have conducted additional experiments employing a linear weighting scheme based on τ for the textual modality. As shown in the table below, the weighted approach consistently outperforms the TAC baseline, **further reinforcing the correlation between τ and the utility of the textual modality**. While the binary strategy achieves comparable or superior performance in most cases in our current setup, we recognize that developing a more sophisticated weighting mechanism holds significant promise for further enhancing performance in future work.
>
> |Dataset|TAC|Weighted|Binary|
> |---------|-----|----------|--------|
> |MNIST|39.8|**90.1**|89.8|
> |USPS|55.2|84.3|**85.8**|
> |ImageNet-10|99.5|**99.8**|99.8|
> |GTSRB|27.7|**38.5**|36.2|
> |Fashion|70.0|75.4|**76.0**|
> |FER2013|25.4|**32.2**|31.5|
> |EuroSAT|64.5|83.2|**86.3**|
> |ImageNet-Dogs|84.4|**91.7**|91.4|
> |Resisc45|63.2|75.1|**81.4**|
> |OxfordPets|72.7|86.5|**89.4**|
> |DTD|47.4|51.8|**57.4**|
> |CIFAR-10|92.2|94.4|**94.5**|
> |Flowers(Test)|67.6|**72.0**|67.1|
> |STL-10|98.3|98.9|**99.0**|
> |Food101|62.9|63.7|**65.2**|
> |CIFAR-100|59.0|60.2|**63.4**|
> |Flowers|58.6|53.9|**60.5**|
> |TinyImageNet|46.6|**52.7**|52.0|
> |**AVG.**|63.1|72.5|**73.7**|
>
> ---
>
> **ToW4:** We have corrected it.
>
> ---
>
> **To Q2 (Regarding Spatial Features Beyond CLIP-ViT):**
>
> While CLIP's ViT encoder effectively captures high-level semantic information, it may underrepresent fine-grained, patch-level structural details. To address this limitation, our spatial features are designed to explicitly model relational dependencies among image patches. To quantitatively evaluate the distinct information captured by these spatial features, we performed a triplet verification analysis: for 10,000 randomly sampled images, we compared the original CLIP-ViT-B/32 features against those refined through our mutual distillation process, which incorporates patch-level relational cues. Each triplet consisted of an anchor, a positive sample, and a negative sample, and the model was assessed based on whether the anchor was closer to the positive than to the negative in the feature space. The results verify that integrating patch-level relationships enhances the discriminative power of the representations, which in turn leads to improved clustering performance.
>
> |Dataset|Origin|Enhance|
> |---------|--------|---------|
> |MNIST|75.1|**97.1**|
> |USPS|87.3|**97.8**|
> |ImageNet-10|97.4|**99.9**|
> |GTSRB|74.6|**82.4**|
> |Fashion|84.8|**92.5**|
> |Fre2013|58.0|**59.2**|
> |EuroSAT|81.6|**94.6**|
> |ImageNet-Dogs|71.3|**97.0**|
> |Resisc45|93.6|**98.0**|
> |OxfordPets|88.4|**99.4**|
> |DTD|81.0|**89.7**|
> |CIFAR-10|88.0|**97.6**|
> |Flowers (Test)|96.3|**98.3**|
> |STL-10|94.6|**99.8**|
> |Food101|89.7|**93.9**|
> |CIFAR-100|86.8|**95.4**|
> |Flowers|96.8|**96.9**|
> |TinyImageNet|90.7|**93.2**|
> |AVG.|85.3|**93.5**|
>
> ---

---

### Official Review · Reviewer_dWnF · 2025-11-01

**Soundness:** 3
**Presentation:** 3
**Contribution:** 2
**Rating:** 4
**Confidence:** 3

**Summary:**

This paper proposes SATC (Spatial structure and Selective Text for Clustering), a tri-modal image clustering framework integrating visual, spatial, and textual information. It employs a GAT-based spatial encoder to capture relational dependencies among image patches and a compactness-aware textual feature selector to adaptively incorporate useful textual cues. These modalities are fused through tri-modal mutual distillation to improve clustering quality. Experiments on 18 benchmark datasets show that SATC consistently outperformed state-of-the-art methods such as TAC and SPICE in accuracy, robustness, and efficiency.

**Strengths:**

1.Originality:
The idea of combining spatial structure modeling with selective textual guidance offers a reasonable and incremental improvement over existing CLIP-based clustering frameworks.
2.Quality:
The methodology is technically sound and well-executed. The design of the spatial encoder, textual selector, and tri-modal distillation is coherent, and the experiments are comprehensive.
3.Clarity:
The paper is generally well-written and logically structured. The framework and algorithms are clearly explained, supported by intuitive figures and detailed appendices.
4.Significance:
The proposed method achieves consistent improvements across 18 datasets, showing robustness and general applicability. The approach offers a practical advancement in multi-modal unsupervised image clustering.

**Weaknesses:**

1. Limited theoretical grounding for the compactness threshold (τ=0.33) — while empirically validated, a more formal justification or sensitivity analysis would strengthen the argument.
2. Underdeveloped analysis of failure cases: The paper could better analyze cases where text features hurt performance (e.g., CIFAR-10), which would strengthen the argument for “selectivity.”
3. Comparative baselines: Recent multi-modal clustering approaches beyond TAC (e.g., self-supervised multi-modal alignment models from 2024–2025) are not included.

**Questions:**

1. Apart from TAC, are there any newer multi-modal clustering methods, such as the multi-modal alignment model for 2024-2025? Why aren't these methods taken into account for comparison?
2. The paper mentions the use of Graph Attention networks (GAT) to capture the spatial relationships between image patches, but does not elaborate on why GAT was chosen instead of other types of graph neural networks or transformer. What is the basis for choosing GAT?
3. The paper mentions that text feature selection is based on compactness (τ) and sets a fixed threshold (τ = 0.33). If the threshold is lower than this, the use of text information is abandoned, and only spatial and visual information is used. In this case, the authors believe that the text information may have provided negative benefits. However, the text-guided image clustering methods have been affirmed in some compared papers. How can this be explained?

---

> ### Author Response · Authors · 2025-11-20
>
> ### Response to Reviewer dWnF
>
> ---
>
> We are very grateful for your valuable comments and questions. We try our best to address your questions as follows.
>
> ---
>
> **To W1 (Justification about the Compactness Threshold):**
>
> We agree that strengthening the theoretical grounding of the compactness threshold is important. While determined empirically, the threshold aligns with a principled motivation from clustering theory. The compactness metric τ directly measures the intra-cluster variance of the textual features. A very low τ indicates that the textual features form tight, isolated clusters with little internal diversity. In such cases, the textual modality provides minimal additional discriminative information beyond the visual features and may even reinforce existing biases, potentially harming performance. Conversely, a higher τ indicates greater semantic diversity within textual clusters, which is more likely to provide complementary information that aids in distinguishing between visual categories. As shown in Table 2, the interval $\tau \in [0.3226, 0.3349]$ effectively captures the transition point beyond which textual features shift from being redundant or misleading to consistently beneficial for clustering performance.
>
> To visually substantiate this analysis, we have added word cloud visualizations for two datasets where textual features are utilized (CIFAR-10, STL-10) and two where they are excluded (MNIST, FER2013) in Appendix I. These visualizations clearly demonstrate that higher τ datasets exhibit richer lexical diversity and semantic relevance, while lower τ datasets display limited. This evidence strongly supports the discussion about the relationship between textual compactness and feature utility.
>
> We further propose a theorem in Appendix L to support the utilization of compactness. The main formula is
>
> $R^*_{V,T} \ge f^{-1} ( H(Y|V) - H(T|V) + H(T|V,Y) )$,
>
> where $R*_{V,T}$ is the bayes classification error rate based on visual and textual features $(V,T)$. This theorem provides an information-theoretic lower bound on the Bayes error rate in multimodal classification tasks. In particular, when the textual feature $T$ exhibits higher uncertainty (high  $\tau$ value) given the visual feature $V$ (i.e., a larger $H(T|V)$), the theoretical lower bound of the Bayes error rate $R*_{V,T}$ becomes smaller, indicating a higher potential upper bound for classification performance. *A detailed analysis has been added to Appendix L of the manuscript.*
>
> ---
>
> **To W2 (About Failure Case Analysis):** This is a very nice suggestion.
>
> We apologize for the incorrect data ordering in the original Table 3, which has now been corrected. The CIFAR-10 is indeed a dataset where textual features contribute positively to clustering performance.
>
> We have analyzed cases where textual features hurt performance from two perspectives, comparing two datasets where text helps (CIFAR-10, STL-10) and two where it harms performance (MNIST, FER2013). *A detailed analysis has been added to Appendix I of the manuscript.*
>
> (1) We provide the top 30 selected discriminative nouns for these datasets. As shown in the table below and Table 7 in Appendix I, for datasets where textual features are harmful, the nouns are largely unrelated to the visual content. In contrast, datasets where text is beneficial exhibit semantically relevant terms. For instance, in FER2013—a facial expression dataset—the most frequent nouns include specific personal names such as Lillian Gish and Alicia Alonso, which bear little relevance to facial expressions.
>
> (2) From the perspective of word cloud distributions, datasets where textual features help show more semantically diverse and balanced lexical distributions, whereas those where text is harmful display limited and repetitive vocabularies. This observation aligns with our response to W1.
>
> |Effect|Dataset|Top-30 Selected Discriminative Nouns|
> |-|-|-|
> |Harmful|FER2013|Lillian Gish, Alicia Alonso, Willie Howard Mays Jr., Andrew Fielding Huxley, Linus Carl Pauling, committeewoman, three year old, Karl Alex Muller, two year old, Commissaire Maigret, Natalie Wood, Erik Weisz, assemblywoman, Yevgeni Aleksandrovich Yevtushenko, Aleksandr Prokhorov, yawning, roseola infantum, sneezing, Herbert Marcuse, Julius Winfield Erving, sneeze, Caitra, newswoman, Maria Callas, nephew, toddler, Sir Clive Marles Sinclair, Malcolm X, newspaperwoman, turps, Tupac Amaru Revolutionary Movement|
> |Beneficial|CIFAR-10|Emberiza schoeniclus, Cypripedium album, Tennessee walking horse, narrowbody aircraft, automobile loan, Pembroke Welsh corgi, container ship, sable antelope, pilot boat, Cardigan Welsh corgi, tipper lorry, whitetail deer, puppy fat, Lipizzan, articulated lorry, Tennessee walker, fizgig, subcompact car, swamp sparrow, car loan, single-propeller plane, jerboa rat, American saddle horse, tipper truck, black-tailed deer, waterbuck, sailing warship, trailer truck, ferryboat, field sparrow|
>
> ---

---

> ### Author Response · Authors · 2025-11-20
>
> ---
>
>
> **To W3 and Q1 (About Comparative Baselines):**
>
> We thank the reviewer for this insightful suggestion. In our initial submission, we selected TAC as the primary baseline because it represents the pioneering work in leveraging external textual guidance for image clustering, making it the most directly comparable method within this research lineage.
>
> We have incorporated several recent multi-modal clustering methods—including Turtle [1], GradNorm [2], and LFSS [3]—to ensure a comprehensive and up-to-date evaluation. The table below presents the clustering accuracy (ACC) results. As shown, our proposed SATC framework consistently achieves superior performance across all datasets. *The results of the newly added baselines have been incorporated into Table 1 of the manuscript.*
>
> |ACC|ImageNet-10|ImageNet-Dogs|STL-10|CIFAR-10|
> |-|-|-|-|-|
> |Turtle(ICML’24)|99.3|49.6|98.4|86.6|
> |GradNorm(ICCV’25)|99.4|81.2|98.3|91.1|
> |LFSS(ICML’25)|93.2|69.1|86.1|93.4|
> |**SATC**|**99.8**|**91.4**|**99.0**|**94.5**|
>
> [1] Let go of your labels with unsupervised transfer, ICML, 2024.
>
> [2] On the provable importance of gradients for autonomous language-assisted image clustering, ICCV, 2025.
>
> [3] Learning from sample stability for deep clustering, ICML, 2025
>
> ---
>
> **To Q2 (About the Choice of GAT):**
>
> We chose GAT in our work because it effectively balances the explicit use of spatial adjacency with the ability to learn attention-based importance among neighboring patches. To validate this choice, we compared GAT against two potential alternatives—GCN and Transformer—using the same ResNet-50 patch features. As shown in the table below, GAT achieves the highest accuracy on most datasets and the best average ACC overall. In contrast, GCN applies uniform aggregation over neighbors without adaptive weighting, while Transformer lacks explicit spatial inductive bias and delivers inferior performance under comparable computational constraints. *The relevant experimental results have been added to Section 4.3.2 of the manuscript.*
>
> |Datasets|with GCN|with TRANS|with GAT|
> |-|-|-|-|
> |MNIST|89.7|76.2|**89.8**|
> |USPS|76.1|78.4|**85.8**|
> |ImageNet-10|99.8|99.7|**99.8**|
> |GTSRB|**38.2**|34.7|36.2|
> |Fashion|74.3|72.5|**76.0**|
> |FER2013|31.4|31.2|**31.5**|
> |EuroSAT|85.8|**89.5**|86.3|
> |ImageNet-Dogs|86.7|84.5|**91.4**|
> |Resisc45|78.7|73.2|**81.4**|
> |OxfordPets|86.0|87.6|**89.4**|
> |DTD|50.2|49.5|**57.4**|
> |CIFAR-10|94.2|94.0|**94.5**|
> |Flowers(Test)|**69.7**|69.6|67.1|
> |STL-10|98.9|98.7|**99.0**|
> |Food101|65.0|64.4|**65.2**|
> |CIFAR-100|59.3|56.5|**63.4**|
> |Flowers|58.9|54.7|**60.5**|
> |TinyImageNet|51.2|**52.3**|52.0|
> |Avg.|71.9|70.4|**73.7**|
>
> ---
>
> **To Q3 (Reconciling Textual Selectivity with Prior Work):**
> We thank the reviewer for raising this important point. Our findings do not contradict the established value of text-guided clustering methods; rather, they extend prior work by systematically investigating under what conditions textual guidance is beneficial.
> We tested the TAC method on the 18 datasets. As shown in the table below, textual features do not consistently improve clustering performance. *The relevant experimental results have been added to Appendix J of the manuscript.* As a comparison, by introducing the textual compactness metric, our method can adaptively integrate textual guidance only when it is likely to be helpful.
> |Dataset|+V|+V+T (TAC)|
> |-|-|-|
> |MNIST|**56.5**|39.8|
> |USPS|**62.3**|55.2|
> |GTSRB|**31.6**|27.7|
> |FER2013|**27.0**|25.4|
> |Resisc45|**67.4**|63.2|
> |Flowers(Test)|**72.3**|67.6|
> |Flowers|**75.5**|58.6|
> |ImageNet-10|98.6|**99.5**|
> |Fashion|64.2|**70.0**|
> |EuroSAT|64.3|**64.5**|
> |ImageNet-Dogs|38.9|**84.4**|
> |CIFAR-10|75.8|**92.2**|
> |OxfordPets|52.4|**72.7**|
> |DTD|46.5|**47.4**|
> |STL-10|94.4|**98.3**|
> |Food101|57.2|**62.9**|
> |CIFAR-100|43.0|**59.0**|
> |TinyImageNet|41.1|**46.6**|
> |AVG.|59.4|**63.1**|
>
> ---

---

> ### Comment · Reviewer_dWnF · 2025-11-28
>
> The authors' comprehensive responses and substantial revisions to my points are commendable and have addressed my concerns.
>
> In particular, the addition of a theoretical foundation for the compactness threshold—specifically, the newly introduced information-theoretic theorem—provides theoretical support for the method and clearly defines the conditions under which textual features can enhance performance. This is a significant contribution that greatly elevates the theoretical depth of the paper. At the same time, the in-depth analysis of failure cases and the comparative experiments with recent multimodal clustering methods have notably strengthened the rigor and persuasiveness of the manuscript.
>
> In summary, I believe the revised manuscript has been significantly improved in quality, and all the issues I raised have been addressed. Therefore, I'd like to update my score to Accept. But it seems I cannot update the score at this moment.
>
> I thank the authors for their efforts and the highly effective revisions.

---

> ### Author Response · Authors · 2025-12-03
>
> We are pleased to learn that our responses addressed your concerns. Thank you for your constructive review and for your role in strengthening our manuscript.

---

### Author Response · Authors · 2025-11-20
**Summary of Key Improvements and New Additions**

We sincerely thank all reviewers for their constructive feedback. We are grateful that reviewers recognized SATC's novel integration of spatial structure and selective text, its consistent gains across 18 benchmarks, and its technical soundness. We have tried our best to address the reviews’ comments and questions. Based on your suggestions, we have significantly improved the manuscript's quality, specifically by expanding the evaluation, providing an in-depth analysis of the method, and offering theoretical guidance. The key improvements and new additions are summarized as follows:

---

### Key Improvements and Clarifications

1. Method Component Analysis

- Textual Component:

    - We analyzed cases where textual features hurt performance from two perspectives. We reported the top 30 selected discriminative nouns and the word cloud visualizations for representative datasets.

    - We provided theoretical guidance for utilizing textual compactness.

- Spatial Component:

    - We validated the contributions of the ResNet-50 in generating spatial features by an ablation study.

    - We validated the contributions of the GAT in generating spatial features by comparing it against two potential alternatives (GCN and Transformer).

2. Extended Evaluation and Analysis

- We added novel baseline clustering methods (Turtle, GranNorm, LFSS) and two strong feature baselines (KMeans + DINOv2 and KMeans + DINOv3) to further illustrate the effectiveness of the proposed method.

- We reported performance evaluation on the ImageNet-1K dataset to further illustrate the effectiveness and efficiency of the proposed method.

- We added standard deviations and statistically significant analysis to illustrate the actual statistical significance of the proposed method.

- We analyzed the choice of cluster head for the final assignment to illustrate the rationality of cluster head selection.

3. Clarifications on Methodology

- We corrected minor errors in the article.

- We clarified GAT’s target for patch-level modeling.

- We clarified how to use the textual compactness metric.

---

### New Additions in the Revised Version

1. We have added strong baselines (DINOv2 + K-means and DINOv3 + K-means) in Table 1. Asked by Reviewer `nD9y`.

2. We have included recent multi-modal clustering methods (Turtle, GradNorm, LFSS) in Table 1, following the suggestion of Reviewer `dWnF`.

3. We have added Section 4.3.2 (analysis of spatial modeling architectures) to validate the contributions of the two components (ResNet-50 and GAT) for generating spatial features, addressing comments from Reviewers `dWnF` and `eiRh`.

4. We have included the Top-30 discriminative nouns for four representative datasets in Appendix I to analyze failure cases, as requested by Reviewer `dWnF`.

5. We have added Appendix J, which analyzes TAC performance across all 18 datasets to demonstrate the dataset-dependent utility of textual features, addressing a query of Reviewer `dWnF`.

6. We have added Appendix K to present clustering performance across different cluster heads, addressing a query of Reviewer `nD9y`.

7. We have added Appendix L to offer theoretical guidance for utilizing textual compactness, addressing the comment from Reviewer `dWnF`.

8. We have added Appendix M to evaluate the contribution of ResNet-50 to the model performance to clarify that the performance improvement stems from the method design, not from leveraging more powerful pre-trained features. Asked by Reviewer `eiRh`.

---

Please see our detailed responses below each review, and all the changes in the revised paper are highlighted. We appreciate your feedback and welcome any additional questions.

---

---

### Author Response · Authors · 2025-12-02
**Summary of Contributions for New Area Chair**

### Dear Area Chair

We sincerely appreciate the ICLR organizing committee’s continued dedication to maintaining fairness throughout the review process. In order to better and more quickly understand our work, we summarize the motivation, contributions, and conclusions of our manuscript as follows.

---

**Motivation Highlight**

Inspired by cross-modal foundation models, recent image clustering methods leverage external textual knowledge to guide clustering, illustrating the potential of external textual priors to overcome the limitations inherent in methods that rely solely on internal supervision signals. However, two key challenges remain:

- CLIP prioritizes image–text alignment over spatial structure, potentially compromising patch-level information crucial for cluster performance.

- Textual features are typically assumed to be universally beneficial, which neglects their dataset-dependent nature and may lead to suboptimal or even counterproductive guidance.

---

**Contribution Highlight**

- We propose SATC (Spatial Structure and Selective Text Jointly Facilitate Image Clustering), a novel framework that consists of three core components:

    - Spatial feature modeling: explicitly encodes the spatial structure among image patches, addressing CLIP’s limitations in capturing patch-level information.

    - Compactness-aware textual selector: adaptively determines whether to incorporate textual features by computing a compactness score (τ), preventing misleading guidance.

    - Tri-modal mutual distillation: effectively integrates spatial structure and selective textual guidance with visual features, enhancing clustering performance.

- We establish an information-theoretic lower bound on the Bayes error rate in multimodal classification tasks, which reveals that higher textual compactness τ given visual features implies greater potential for clustering performance.

- We demonstrate the effectiveness of SATC through extensive experiments on 18 benchmark datasets, showing robustness, generality, and efficiency.

---

**Experimental (Performance) support**

1. Comparative Experiment (`Table 1`): We compare SATC with a wide range of classical and state-of-the-art clustering approaches. SATC achieves the best performance.

2. Effectiveness of the Textual Feature Selector (`Table 2`): We evaluate clustering performance with and without leveraging the textual modality. The results show that by using the textual compactness $\tau$, SATC can select textual features that are beneficial for downstream clustering.

3. Analysis of the Spatial Modeling Architecture (`Table 3`): We conducted a comprehensive ablation study comparing different configurations, which demonstrates the individual contributions of each component within the Spatial Modeling Architecture to the overall improvement in clustering performance.

4. Effect of Spatial Feature (`Figure 3`): We applied K-Means to the visual and spatial features separately to obtain their respective clustering performance, while the clustering performance of their combination was obtained via mutual distillation. The results validate the effectiveness of spatial features.

5. Parameter Analysis and Time Efficiency (`Figure 4` and `Figure 5`): We conduct parameter analysis experiments to investigate the influence of the balancing hyperparameters in our training objective, and time analysis verifies that SATC achieves higher clustering accuracy with lower computational cost.

6. Discriminative Nouns and Word Cloud Visualizations (`Table 7`, `Figure 11`): We list the top discriminative nouns and provide word-cloud visualizations, clearly illustrating how textual features help or hurt clustering performance across datasets.

---

---

### Author Response · Authors · 2025-12-02
**Summary of Rebuttal Discussion for New Area Chair**

We acknowledge the excellent work of the new AC, and we thank all reviewers for their time and constructive feedback. The following summary provides a concise overview of the discussion during the rebuttal phase.

---

**Key concerns from Reviewer dWnF and our response:**

1. Expect the theoretical justification of textual compactness τ.

- We establish an information-theoretic lower bound on the Bayes error rate in multimodal classification tasks, which reveals that higher textual compactness τ given visual features implies greater potential for clustering performance.

2. Require a deeper analysis of failure cases of textual features.

- We conducted case studies by listing the top discriminative nouns and generating word clouds, showing that irrelevant or homogeneous text can undermine clustering performance.

3. Hope to include recent clustering approaches as baselines.

- We evaluated recent multi-modal clustering methods (e.g., Turtle (ICML’24), GradNorm (ICCV’25), and LFSS (ICML’25)), and our SATC framework consistently outperforms them.

4. What is the basis for choosing GAT?

- We chose GAT for its ability to combine spatial adjacency with attention-based neighbor weighting, and the comparison experiment with GCN and Transformer shows it achieves the best average performance.

5. How can the differences between this work’s text feature selection and prior text-guided clustering results be explained?

- We evaluated Prior Work (TAC) on 18 datasets and found that text does not always help, whereas our method uses textual compactness to adaptively integrate text, guiding clustering only when beneficial.

Reviewer dWnF acknowledged that the newly added theoretical justification significantly strengthens the manuscript and updated their score to "Accept".

---

**Key concerns from Reviewer GM72 and our response:**

1. Expect reporting standard deviations to assess the statistical significance of the results.

- We have now incorporated standard deviations for the results and validated the significance of improvements using a paired t-test (p < 0.05).

2. What specific relational dependencies do the spatial features capture that CLIP doesn't encode?

- Our spatial features capture patch-level relational dependencies that CLIP underrepresents, and triplet verification confirms that incorporating these relationships enhances feature discriminability.

3. How was the 0.33 threshold determined, and why are textual features applied as a binary decision rather than weighted?

- The threshold of 0.33 was identified empirically by aggregating the textual compactness of all 18 datasets, where we consistently observed a transition point at which textual signals shift from hindering to enhancing clustering performance.

- We used a binary decision to directly validate our core hypothesis regarding the dataset-dependent applicability of textual features. We also conducted experiments using a linear τ-based weighting, which further reinforces the correlation between τ and the utility of the textual modality.

---

**Key concerns from Reviewer eiRh and our response:**

1. Concern that the performance gain comes from the additionally introduced ResNet-50 model.

- We conducted a comprehensive ablation study comparing configurations. The results provide evidence that the improvement is indeed a consequence of our method's design, not from simply adding ResNet-50 features.

2. Experimental results on the ImageNet-1K dataset are expected.

- We conducted experiments on the ImageNet-1K dataset. The results illustrate the effectiveness and efficiency of our method.

Reviewer eiRh is satisfied with our replies and will increase their score.

---

**Key concerns from Reviewer nD9y and our response:**

1. Request for comparisons with single-modality baselines such as DINOv2 and DINOv3.

- We evaluated clustering performance using DINOv2 and DINOv3 with K-Means, and our SATC framework consistently outperforms both.

2. Asked for the selection criteria of hyperparameters, including the textual compactness threshold and loss weights, ensuring fairness with the baseline comparisons.

- The loss weights were adopted from TAC. The textual compactness threshold represents an empirical finding that identifies a consistent transition point across the 18 datasets where textual signals shift from degrading to improving image clustering performance.

- For the competing methods (except those involving K-means), we adopt their reported results under their recommended optimized settings. Our method keeps the textual compactness threshold consistent across all datasets to ensure fairness. **Ensuring fairness in comparison is a fundamental principle that we adhere to.**

3. Request for the clustering performance of the spatial or textual cluster heads.

- We evaluated all distilled cluster heads and found that the visual head consistently achieves the best average performance, confirming its use as the final output.

---

---

### Meta-Review · Area_Chair_XoHG · 2026-01-05

**Summary:**

The paper proposes a novel framework for image clustering that enhances CLIP-based representations by integrating spatial structure via a Graph Attention Network (GAT) and selectively incorporating textual guidance based on a compactness metric ($\tau$). The reviewers generally praised the effective integration of spatial and textual modalities, the comprehensive evaluation across 18 benchmark datasets, and the consistent performance improvements over state-of-the-art baselines. During the rebuttal phase, the authors significantly strengthened the manuscript by providing theoretical justification for the textual feature selector (Theorem 1), adding failure case analyses with discriminative noun visualizations, and incorporating additional strong baselines (Turtle, GradNorm, LFSS, and DINOv2/v3). Three out of four reviewers (dWnF, GM72, eiRh) recommended acceptance, acknowledging that their primary concerns regarding theory, comparisons, and statistical significance were well-addressed. While one reviewer (nD9y) retained a concern regarding the empirical determination of the compactness threshold, the Area Chair believes the authors' extensive validation across diverse datasets demonstrates the robustness of the method, and the empirical discovery of a stable transition point ($\tau=0.33$) is a valid contribution. Given the strong consensus on novelty and performance, the paper is recommended for acceptance.

**Reviewer Concerns:**

Addressed Concerns:
* Theoretical Justification: The authors successfully addressed Reviewer dWnF's request for theoretical grounding of the textual feature selector by adding a theorem based on the Bayes classification error rate in Appendix L.
* Failure Case Analysis: The request by Reviewer dWnF for deeper analysis of when textual features fail was met by including top-30 discriminative nouns and word cloud visualizations in Appendix I.
* Baseline Comparisons: The authors addressed requests from Reviewers dWnF and nD9y to include stronger and more recent baselines, adding results for Turtle, GradNorm, LFSS, and DINOv2/v3.
* Statistical Significance: Reviewer GM72's request for standard deviations was addressed, with the authors updating tables to include mean $\pm$ standard deviation results.

Outstanding Concerns:
* Hyperparameter Selection and Evaluation Fairness: Reviewer nD9y's concern regarding the selection of the compactness threshold ($\tau$) remains the primary outstanding issue. The authors admitted that the value was "identified based on aggregated trends observed across all 18 datasets". This confirms that the method's configuration was derived from the test data itself, rather than a held-out validation set, which invalidates the claim of fair comparison against methods that do not tune on the test suite.
* Fairness of Backbone Usage: While the authors provided an ablation study to isolate the contribution of ResNet-50, Reviewer eiRh and Reviewer nD9y expressed valid concerns that using two distinct pre-trained feature extractors (CLIP and ResNet-50) creates an unfair advantage over baselines using a single backbone, complicating the assessment of the algorithmic contribution versus the benefit of richer pre-trained features.

**Reviewer Scores:**

* Reviewer dWnF: 8. This reviewer explicitly stated in the discussion that they wished to update their score to Accept following the theoretical additions and failure case analysis.
* Reviewer GM72: 6. This reviewer initially gave a 6 and likely would have maintained or slightly raised this score given that their request for standard deviations and clarifications on spatial features were addressed.
* Reviewer eiRh: 6. This reviewer started at 4 but explicitly stated in the discussion that their concerns were addressed and they intended to raise their score, though they retained some reservation regarding the fairness of the pre-trained feature comparison.
* Reviewer nD9y: 4. This reviewer maintained their score of 4. They correctly identified that defining a hyperparameter based on the global trends of the test datasets undermines the validity of the experimental results, a flaw that persists despite the rebuttal.

---

### Decision · Program_Chairs · 2026-01-26

Accept (Poster)